# Bridging Spherical Black-Box Optimizers

**Johannes Ackermann** [1]  **Stefano Peluchetti** [2]

## Abstract

When gradient information is unavailable, black-box optimization (BBO) methods provide a practical alternative. While Evolution Strategies (ES), Consensus-Based Optimization (CBO), Optimization via Integration (OVI), and related methods have each been studied independently, their connections remain underexplored. We unify these approaches within a common theoretical framework, revealing that they differ primarily in two design choices: fitness aggregation (controlling sharpness preference) and consensus scope (controlling modality). Leveraging these insights, we introduce hybrid optimizers that interpolate between existing methods. Our ES-OVI hybrid allows explicit control over the preference for flat minima, enabling a trade-off between performance and robustness in continuous control tasks. Our CBO-OVI hybrids combine the higher-dimensional efficiency of parametric methods with the multimodal capabilities of particle-based approaches, achieving competitive results on language model merging under limited evaluation budgets. We validate our methods on standard BBO benchmarks and higher-dimensional locomotion tasks, demonstrating that the hybrid methods can outperform their constituent algorithms.

## 1. Introduction

Stochastic gradient descent and its variants have become the dominant optimizers for deep learning models. However, gradients may be unavailable or not useful for training (Metz et al., 2022), for example when optimizing through simulators, using external APIs, or in Reinforcement Learning. In such cases, zeroth-order or black-box optimizers offer a solution as they only require objective function evaluations.

JA did his work while interning at Sakana AI. [1]The University of Tokyo [2]Sakana AI. Correspondence to: Johannes Ackermann <ackermann@ms.k.u-tokyo.ac.jp>.

*Proceedings of the $43^{rd}$ International Conference on Machine Learning*, Seoul, South Korea. PMLR 306, 2026. Copyright 2026 by the author(s).

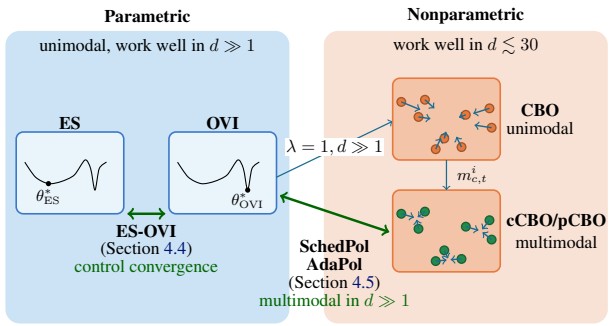

*Figure 1.* We investigate connections between the parametric ES, OVI, the nonparametric CBO, and further related optimizers. By utilizing these connections, we can derive hybrid methods, indicated by **green** arrows, that combine the strengths of existing optimizers: ES-OVI allows us to control convergence characteristics, SchedPol and AdaPol combine CBO and OVI updates, allowing us to obtain multiple optima in higher-dimensional tasks.

There are two main families of black-box optimizers, which are usually treated separately: Parametric and non-parametric methods. On the parametric side, a prominent family are Evolution Strategies (ES) (Rechenberg, 1973). Within ES, we specifically focus on Natural ES (NES) (Wierstra et al., 2008), which maintain a parametric search distribution, updating parameters based on the expectation of the objective over sampled points. Optimization via Integration (OVI) (Andrieu et al., 2024) is also parametric, but proposes to instead repeatedly tilt a search distribution by the objective values. On the non-parametric side, particle-based approaches such as Consensus Based Optimization (CBO) (Pinnau et al., 2017) evolve a population of candidate solutions. This is done based on the function value of all particles, or in the case of Polarized CBO (pCBO) (Bungert et al., 2025) only based on the value of nearby particles, allowing us to obtain multiple optima.

While ES, OVI, and CBO have each been linked to gradient descent (Salimans et al., 2017; Riedl et al., 2023; Andrieu et al., 2024), the connections between distribution-update methods (ES/OVI) and CBO particle methods have, to our knowledge, not been investigated. In this work, we propose a unified framework (Section 2) for these optimizers: a single master update equation (MU) from which ES, OVI, CBO, and their variants emerge as special cases, differing primarily in their fitness aggregation and interaction scope

Table 1. Instantiations of the master update (MU), unifying all considered BBO methods.

| Method | $\Psi(F)$ (sharpness) | $K_t^{i,j}$ (scope / interaction) | $(\mu_t, \lambda_t)$ (transport) | $\sigma_t s(x - m)$ (noise) |
|---|---|---|---|---|
| **CH / OVI** | $\exp\{-\beta F\}$ | 1 (global) | $(0, 1)$ | $\sigma$ |
| **ES**[†] | $1 - \frac{\eta}{\sigma^2}\left(F - \overline{F}\right)$ | 1 (global) | $(0, 1)$ | $\sigma$ |
| **CBO** | $\exp\{-\beta F\}$ | 1 (global) | $(1 - \lambda, \lambda)$ | $\sigma\|x - m\|$ |
| **pCBO** | $\exp\{-\beta F\}$ | $\exp\{-\|x_t^i - x_t^j\|^2/(2\kappa^2)\}$ | $(1 - \lambda, \lambda)$ | $\sigma\|x - m^i\|$ |
| **cCBO**[‡] | $\exp\{-\beta F\}$ | $\sum_{c=1}^{N_C} \frac{p_t^{i,c} p_t^{j,c}}{Z_t^c}$ | $(1 - \lambda, \lambda)$ | $\sigma\|x - m^i\|$ |
| **DE** | $\exp\{-\beta F\}$ | $\exp\{-\|x_t^i - \sqrt{\alpha_t}x_t^j\|^2/(2(1-\alpha_t))\}$ | $(\mu_t^{\mathrm{DDIM}}, \lambda_t^{\mathrm{DDIM}})$ | $\sigma_t^{\mathrm{DDIM}}$ |

ES[†]: with antithetic sampling; $\Psi$ depends on $\overline{F}$ as well (population mean of loss values).
cCBO[‡]: $p_t^{i,c}$ are cCBO assignments (4), $Z_t^c = \sum_{j=1}^N p_t^{j,c} \exp\{-\beta F(x_t^j)\}$.

(Table 1). We then leverage these insights to derive new optimizers that address the weaknesses of each method. We provide an overview in Figure 1. We empirically evaluate their convergence characteristics, and evaluate their performance on low-dimensional BBO benchmarks and higher-dimensional locomotion tasks. Finally, we investigate their application to evolutionary model merging under limited evaluation budgets. We propose to address this setting as a multi-modal optimization problem, in which our proposed CBO-OVI combinations are successful.

## 2. Unifying Framework

We can view many black-box optimizers as iterating two steps: (i) compute a (possibly local) fitness-weighted consensus point, and (ii) update particles by (partially) moving toward this consensus while injecting exploration noise. This yields a single master update that recovers ES, OVI, CH, CBO, pCBO, clustered CBO (cCBO), and Diffusion Evolution (DE).

**Master Update (MU)**  Given particles $x_t^1, \ldots, x_t^N \in \mathbb{R}^d$, define

$$x_{t+1}^i = \mu_t x_t^i + \lambda_t m_t^i + \sigma_t s(x_t^i - m_t^i)\epsilon_t^i, \quad \epsilon_t^i \overset{\text{i.i.d.}}{\sim} \mathcal{N}(0, I),$$

$$m_t^i = \sum_{j=1}^N w_t^{i,j} x_t^j, \qquad w_t^{i,j} = \frac{a_t^{i,j}}{\sum_{\ell=1}^N a_t^{i,\ell}},$$

$$a_t^{i,j} = \Psi(F(x_t^j)) K_t^{i,j}.$$

(MU)

Here $\Psi$ is a fitness transformation (sharpness preference), $K_t^{i,j} \geq 0$ is an interaction matrix controlling how particles influence each other (global vs. local vs. clustered interactions), $\lambda_t$ controls attraction toward consensus, $\mu_t$ controls persistence, $\sigma_t$ controls exploration, and $s(\cdot)$ determines noise scaling.

In all methods aside from cCBO, $K_t^{i,j}$ is constructed from a kernel: $K_t^{i,j} = k_t(x_t^i, x_t^j)$. If $K_t^{i,j} = 1$, then $m_t^i$ does

not depend on $i$, and we obtain a global consensus point $m_t$. Most methods satisfy $\mu_t + \lambda_t = 1$ (convex transport), but we keep $(\mu_t, \lambda_t)$ separate to accommodate the DDIM (Song et al., 2021a) sampler in DE. If $\mu_t = 0$ and $\lambda_t = 1$, then (MU) resamples the population around the consensus ($x_{t+1}^i = m_t^i + \sigma_t \epsilon_t^i$), covering OVI/CH and ES.

Table 1 shows how the discussed BBO methods instantiate (MU). For ES, we assume mean-zero perturbations via antithetic sampling, a standard practice (Salimans et al., 2017). Under this assumption, ES can be written exactly in terms of (MU). Without it, the correspondence remains a close approximation for $N \gg 1$. We defer to Appendix A.6 for details. The equivalence of CH and OVI is established in Appendix A.5. Finally, cCBO fits (MU) by using a cluster-induced interaction matrix $K_t$ (Appendix A.7).

## 3. Background

We consider a loss function $F : \mathbb{R}^d \to \mathbb{R}$, which we aim to minimize. In the black-box optimization setting, we can evaluate $F$ at any point, but cannot access its gradient $\nabla F$.

### 3.1. Parametric Methods

**Stochastic Smoothing (SS)**  constructs a differentiable surrogate for non-differentiable or black-box functions (Katkovnik & Kulchitsky, 1972; Spall, 2003; Nesterov & Spokoiny, 2017). Given a smoothing distribution with density $\pi$, the smoothed objective is $J_\pi^{\mathrm{SS}}(\theta) = \mathbb{E}_{\epsilon \sim \pi}[F(\theta + \epsilon)]$. Using the likelihood-ratio estimator, the gradient can be expressed as $\nabla_\theta J_\pi^{\mathrm{SS}}(\theta) = \mathbb{E}_{\epsilon \sim \pi}[F(\theta + \epsilon)\nabla_\epsilon - \log \pi(\epsilon)]$ (see Appendix A.2 for details). For a spherical Gaussian $\pi = \mathcal{N}(0, \sigma^2 I)$, this yields the gradient descent updates

$$\theta_{t+1} = \theta_t - \frac{\eta}{\sigma N} \sum_{i=1}^N F(\theta_t + \sigma \epsilon^i)\epsilon_t^i, \quad \epsilon_t^i \overset{\text{i.i.d.}}{\sim} \mathcal{N}(0, I),$$

(SS)

where $\eta > 0$ is the learning rate.

**Natural Evolution Strategies (NES)** (Wierstra et al., 2008; 2014) optimize a parametric search distribution $\pi(x; \theta)$ to maximize expected fitness $J_\pi^{\text{ES}}(\theta) = \mathbb{E}_{x \sim \pi(\cdot; \theta)}[-F(x)]$. For a Gaussian $\pi = \mathcal{N}(\theta, \sigma^2 I)$ with fixed variance, the NES updates are

$$\theta_{t+1} = \theta_t - \eta \mathbb{E}_{\epsilon \sim \mathcal{N}(0, \sigma^2 I)} [F(\theta_t + \epsilon)\epsilon]. \quad \text{(NES)}$$

For spherical Gaussians, NES and SS yield equivalent updates (Section 4.1); we refer to both as Evolution Strategies (ES). Salimans et al. (2017) demonstrate the effectiveness of ES for reinforcement learning, showing favorable scaling with parallel computation. In practice, fitness shaping transformations (e.g., rank-based) are applied before computing updates, which we discuss more in Appendix A.1.

**Optimization via Integration (OVI)** (Andrieu et al., 2024) replaces the expectation in ES with a "log-sum-exp" aggregation, a relaxation of the minimum. While ES minimizes the expected loss $\mathbb{E}_\pi[F(x)]$, OVI minimizes

$$J^{\text{OVI}}(\theta) = -\log \mathbb{E}_{\epsilon \sim \mathcal{N}(0, \sigma^2 I)} [\exp(-\beta F(\theta + \epsilon))],$$

where $\beta > 0$ controls the sharpness of the minimum approximation. The OVI updates correspond to gradient descent on $J^{\text{OVI}}$ with step size $\sigma^2$:

$$\theta_{t+1} = \theta_t - \sigma^2 \nabla_\theta J^{\text{OVI}}(\theta_t) = \frac{\sum_{i=1}^N x_t^i e^{-\beta F(x_t^i)}}{\sum_{i=1}^N e^{-\beta F(x_t^i)}}, \quad \text{(OVI)}$$

where $x_t^i \overset{\text{i.i.d.}}{\sim} \mathcal{N}(\theta_t, \sigma^2 I)$. This computes a weighted average of samples, with weights $w_t^i \propto \exp(-\beta F(x_t^i))$ favoring low-loss regions. OVI can also be derived from a Bayesian perspective (see Appendix A.3). In practice, Andrieu et al. (2024) set $\beta = 1/\hat{\sigma}_F$, where $\hat{\sigma}_F$ is the sample standard deviation of fitness values.

### 3.2. Nonparametric Methods

**Consensus Based Optimization (CBO)** (Pinnau et al., 2017; Carrillo et al., 2021) is a particle-based method for global optimization. $N$ interacting particles $x_t^1, \ldots, x_t^N \in \mathbb{R}^d$, evolve according to the dynamics specified by the system of SDEs

$$dx_t^i = -\lambda(x_t^i - m_t)dt + \sigma \|x_t^i - m_t\| dW_t^i, \quad (1)$$

where $W_t^i$ are independent Brownian motions and $m_t$ is the consensus point, defined as the weighted average

$$m_t = \frac{\sum_{i=1}^N x_t^i \exp(-\beta F(x_t^i))}{\sum_{i=1}^N \exp(-\beta F(x_t^i))}. \quad (2)$$

The parameter $\beta > 0$ controls the sharpness of the weighting: as $\beta \to \infty$, the consensus point converges to the particle with lowest objective value. The drift term attracts each

particle toward the consensus, while the diffusion term adds exploration noise scaled by the Euclidean distance to the consensus. The Euler-Maruyama discretization of (1) with step size $\Delta t = 1$ yields the CBO updates:

$$x_{t+1}^i \leftarrow x_t^i - \lambda(x_t^i - m_t) + \sigma \|x_t^i - m_t\| \epsilon_t^i, \quad \epsilon_t^i \overset{\text{i.i.d.}}{\sim} \mathcal{N}(0, I). \quad \text{(CBO)}$$

The hyperparameter $\lambda \in (0, 1]$ controls the strength of attraction to the consensus point, while $\sigma$ controls exploration strength. A distinguishing feature of CBO compared to other prior particle-based methods is its amenability to the mean-field limit $N \to \infty$, which allows establishing convergence guarantees to the global minimum under suitable assumptions (Pinnau et al., 2017; Carrillo et al., 2021).

**Polarized CBO (pCBO)** Standard CBO converges to a single global optimum, but in many applications we may prefer multiple good local optima—for instance, when optimizing a surrogate objective (e.g., performance in a simulated environment) while caring about a different metric at deployment. To address this, Bungert et al. (2025) proposed Polarized CBO, which replaces the global consensus point with a local consensus point for each particle. The local consensus point for particle $x_t^i$ at time $t$ is defined as

$$m_{p,t}^i = \frac{\sum_{j=1}^N x_t^j \exp(-\beta F(x_t^j)) k(x_t^i, x_t^j)}{\sum_{j=1}^N \exp(-\beta F(x_t^j)) k(x_t^i, x_t^j)}. \quad (3)$$

The localizing kernel is typically Gaussian: $k(x, y) = \exp(-\|x - y\|^2/(2\kappa^2))$. The bandwidth $\kappa$ controls the effective interaction range between particles. This creates polarization: particles in different regions of the domain effectively form independent subpopulations, each converging to its own local optimum. For $k = 1$, we recover standard CBO. Particles are then updated via the (CBO) updates with $m_{p,t}^i$ replacing the global consensus $m_t$ for each particle $x_t^i$.

**Clustered CBO (cCBO)** While pCBO can find multiple optima in low dimensions, Bungert et al. (2025) show it struggles when $d \geq 10$ due to the curse of dimensionality affecting kernel-based locality. To improve scalability, they propose cCBO, which maintains $N_c$ cluster centers $c^1, \ldots, c^{N_c}$ and soft assignments $p_t^{i,j} \in [0, 1]$ representing the probability that particle $i$ belongs to cluster $j$. Each particle $x_t^i$ is attracted to its individual consensus point, defined as the weighted combination of cluster centers:

$$m_{c,t}^i = \sum_{j=1}^{N_c} p_t^{i,j} c_t^j.$$

Particles are then updated via CBO dynamics (CBO) with $m_{c,t}^i$ replacing the global consensus $m_t$. The cluster assign-

ments and centers are updated as:

$$p_t^{i,j} \leftarrow \frac{r_t^{i,j} k(x_t^i, c_t^j)}{\sum_{j'=1}^{N_c} r_t^{i,j'} k(x_t^i, c_t^{j'})}, \quad r_t^{i,j} = \left( \frac{p_t^{i,j}}{\max_{j'} p_t^{i,j'}} \right)^\alpha$$

$$c_t^j \leftarrow \frac{\sum_{i=1}^{N} x_t^i p_t^{i,j} \exp(-\beta F(x_t^i))}{\sum_{i=1}^{N} p_t^{i,j} \exp(-\beta F(x_t^i))}.$$

$$(4)$$

The parameter $\alpha > 0$ controls assignment sharpness: larger $\alpha$ encourages particles to commit to a single cluster. Each cluster center $c_t^j$ evolves as a consensus point over its assigned particles, enabling simultaneous optimization toward $N_c$ distinct optima.

**Consensus Hopping (CH)**   Consensus Hopping (Riedl et al., 2024) is defined by the update

$$x_{t+1}^i \leftarrow m_t + \sigma' \epsilon_t^i, \quad \epsilon_t^i \sim \mathcal{N}(0, I), \qquad \text{(CH)}$$

where $m_t$ is defined as in (2). This resembles CBO with $\lambda = 1$, but uses a state-independent diffusion coefficient $\sigma'$ rather than the distance-dependent term $\sigma \|x_t^i - m_t\|$. In high dimensions, the radii $\|x_t^i - m_t\|$ concentrate across particles, making the two formulations approximately equivalent, as we discuss in Section 4.2.

**Diffusion Evolution (DE)**   Zhang et al. (2025) proposed Diffusion Evolution (DE), motivated by denoising diffusion models (Sohl-Dickstein et al., 2015; Song et al., 2021b). DE computes a local consensus for each particle using a time-dependent kernel:

$$\hat{x}_0^i = \frac{\sum_{j=1}^{N} x_t^j \exp(-\beta F(x_t^j)) k_t(x_t^i, x_t^j)}{\sum_{j=1}^{N} \exp(-\beta F(x_t^j)) k_t(x_t^i, x_t^j)}, \qquad (5)$$

where the kernel

$$k_t(x, y) = \exp \left( -\frac{\|x - \sqrt{\alpha_t} y\|^2}{2(1 - \alpha_t)} \right) \qquad (6)$$

derives from the diffusion forward process with schedule $\alpha_t$. Particles are updated using the DDIM sampler (Song et al., 2021a):

$$x_{t-1}^i = \sqrt{\alpha_{t-1}} \hat{x}_0^i + \sqrt{1 - \alpha_{t-1} - \sigma_t^2} \cdot \frac{x_t^i - \sqrt{\alpha_t} \hat{x}_0^i}{\sqrt{1 - \alpha_t}} + \sigma_t \epsilon_t^i,$$

$$\text{(DE)}$$

with $\epsilon_t^i \overset{\text{i.i.d.}}{\sim} \mathcal{N}(0, I)$. See Appendix A.4 for the full derivation from the diffusion model perspective.

## 4. Connections

### 4.1. SS ≡ NES

Although Stochastic Smoothing and Natural Evolution Strategies arise from different motivations (SS from constructing a differentiable surrogate, NES from optimizing a

search distribution) they yield identical updates when using *Spherical Gaussian* distributions. Comparing (SS,NES), we see that both reduce to the same gradient estimator, differing only by a $\sigma^2$ scaling absorbed into the learning rate $\eta$. This equivalence justifies referring to both methods simply as ES throughout this work.

### 4.2. CH ≈ CBO

While CH uses state-independent noise $\sigma'$ and CBO uses distance-scaled noise $\sigma \|x_t^i - m_t\|$, these formulations become approximately equivalent in high dimensions. The radii $\|x_t^i - m_t\|$ concentrate across particles (and, in clustered variants, within each cluster) around a common value $r_t = \sqrt{\frac{1}{N} \sum_{i=1}^{N} \|x_t^i - m_t\|^2}$, so $\sigma \|x_t^i - m_t\| \approx \sigma r_t$ for most particles. Consequently, CBO with $\lambda = 1$ behaves similarly to CH with noise scale $\sigma_t' = \sigma r_t$. These heuristic arguments can be made precise by combining the mean-field limit $N \to \infty$ (yielding a particle law $\rho_t$) with the high-dimensional limit $d \to \infty$ (yielding the concentration of the empirical RMS radius $r_t$ toward a deterministic schedule).

### 4.3. DE ≈ pCBO

We show that Diffusion Evolution is more accurately understood as pCBO with a time-dependent kernel. In particular, we note that (5) is *not* a valid approximation to the optimal denoiser $\mathbb{E}_\pi[x_0 \mid x_t]$ (see Appendix A.4 for a brief review of denoising diffusion models). As $N \to \infty$, the MC approximation in (5) converges to

$$\hat{x}_0 \to \frac{\int x_0 \pi(x_0) \rho_t(x_0) q_{\alpha_t}(x_t \mid x_0) \mathrm{d}x_0}{\int \pi(x_0) \rho_t(x_0) q_{\alpha_t}(x_t \mid x_0) \mathrm{d}x_0},$$

where $\rho_t$ is the distribution of (infinitely many) particles at time $t$. Comparing with $\mathbb{E}_\pi[x_0 \mid x_t]$, there is an extra factor $\rho_t$. That is, the denoiser assumes a time-dependent data distribution proportional to $\pi \rho_t$, instead of being proportional to $\pi$. A correct particle-based importance-sampled approximation would require weights $1/\rho_t$, but these are unavailable since $\rho_t$ itself is unknown. The denoised estimate $\hat{x}_0^i$ in (5) is exactly a pCBO local consensus point (3) with kernel (6). The time-dependence of $k_t$ provides an annealing schedule: as $\alpha_t \to 1$, the bandwidth $(1 - \alpha_t) \to 0$, transitioning from global interaction to local interaction.

DE employs the DDIM sampler instead of the Euler-Maruyama scheme typically employed in CBO. Both are valid discretization schemes. Moreover, DE employs a deterministic diffusion schedule, in contrast to (CBO). However, we already noted that in high-dimensional settings the state-dependency of CBO's diffusion coefficient is often limited. In summary, we argue that DE is most easily understood as a time-inhomogeneous implementation of pCBO, with different discretization scheme and noise schedules.

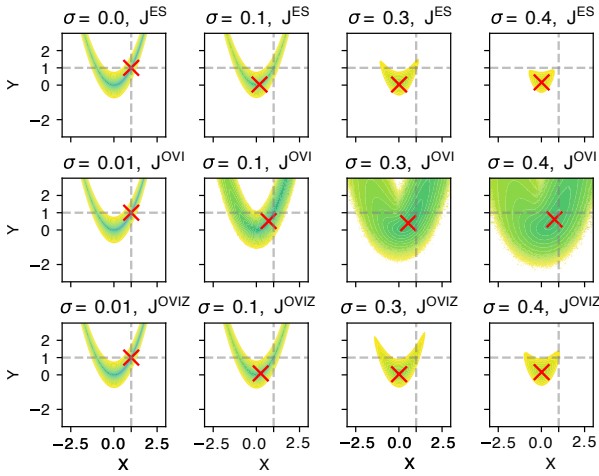

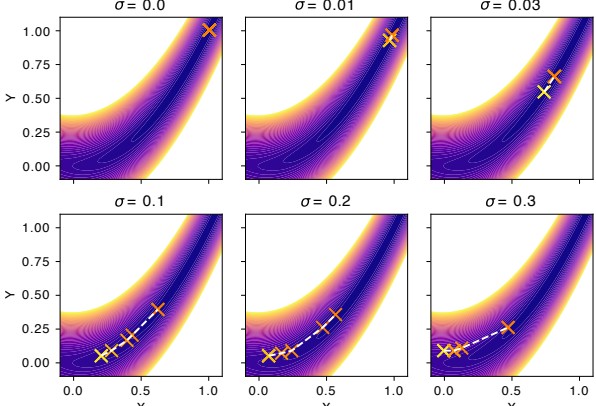

*Figure 3*. **ES-OVI lets us control the flatness of the optimum.** Markers indicate the minimum of $J^\alpha$ on the Rosenbrock function for different values of $\alpha$, yellow ($\alpha = 0$, ES) to red ($\alpha = 1$, OVI).

*Figure 2*. **ES prefers flat basins, OVI sharp optima.** Objectives of ES $J^{\text{ES}}$, OVI $J^{\text{OVI}}$ (with $\beta = 1$), and fitness-scaled OVI (with $\beta = 1/\hat{\sigma}_F$), for different noise levels $\sigma$ on the Rosenbrock function. The true minimum is at $(1, 1)$; the red x marks the minimum of each smoothed objective. Under $J^{\text{ES}}$, the minimum moves toward the flat valley around $[0, 0]$, while it stays closer to the true minimum under $J^{\text{OVI}}$.

### 4.4. ES ↔ OVI

Both ES and OVI use a symmetric Gaussian proposal distribution $\mathcal{N}(\theta, \sigma^2 I)$. Their difference lies in the surrogate objectives $J^{\text{ES}}$ and $J^{\text{OVI}}$:

$$J^{\text{ES}}(\theta) = \mathbb{E}_{\epsilon \sim \mathcal{N}(0, \sigma^2 I)}[F(\theta + \epsilon)]$$
$$J^{\text{OVI}}(\theta) = -\log \mathbb{E}_{\epsilon \sim \mathcal{N}(0, \sigma^2 I)} \left[ \exp(-\beta F(\theta + \epsilon)) \right].$$

At first glance, both objectives appear similar: each samples perturbations from a symmetric Gaussian and aggregates function values. When $F$ is locally flat around $\theta$, the two objectives indeed yield similar landscapes. However, they differ markedly near sharp minima: ES smooths them out, whereas OVI tends to preserve or even accentuate them.

To illustrate, we visualize both surrogate objectives on the Rosenbrock function $F(x, y) = (1 - x)^2 + 100(y - x^2)^2$ (Rosenbrock, 1960), which has a sharp global minimum at $(1, 1)$ and a broad, suboptimal valley near the origin. Figure 2 shows that as $\sigma$ increases, the minimum of $J^{\text{ES}}$ shifts toward the origin where $F(0, 0) = 1$. In contrast, the minimum of $J^{\text{OVI}}$ with $\beta = 1$ remains close to the true optimum, even for large $\sigma$. When using adaptive scaling $\beta = 1/\hat{\sigma}_F$, OVI behaves more similarly to ES: Scaling by the inverse standard deviation attenuates sharp minima (which induce high variance in $F$) while leaving flat regions relatively unchanged.

Flat minima are often preferable to sharp ones, as they are associated with better generalization in both supervised learning (Hochreiter & Schmidhuber, 1997; Foret et al., 2021) and reinforcement learning (Lee & Yoon, 2025). However,

as illustrated by the Rosenbrock example in Figure 2, an excessive preference for flat regions can be detrimental, causing convergence to suboptimal solutions.

**ES-OVI** To allow us to control the convergence behavior of the optimizer depending on the application, we propose to interpolate the two methods via a convex combination of their gradients:

$$\nabla J^\alpha(\theta) = \alpha \nabla J^{\text{OVI}}(\theta) + (1 - \alpha) \nabla J^{\text{ES}}(\theta), \quad \alpha \in [0, 1].$$

Since both objectives use the same samples $\epsilon_t^i$ and function evaluations $F(\theta + \epsilon_t^i)$, this combination requires no additional function evaluations. As shown in Figure 3, varying $\alpha$ smoothly interpolates between the convergence points of ES ($\alpha = 0$) and OVI ($\alpha = 1$), allowing us to obtain a solution with a desired flatness. We refer to this method as ES-OVI.

### 4.5. OVI ≡ CH

While nonparametric CBO and parametric OVI may appear quite dissimilar, we observe that Consensus Hopping is equivalent to OVI. Expanding the (CH) update with the definition of the consensus point yields

$$x_{t+1}^i = \frac{\sum_{j=1}^N x_t^j \exp(-\beta F(x_t^j))}{\sum_{j=1}^N \exp(-\beta F(x_t^j))} + \sigma' \epsilon_t^i,$$

which is precisely the (OVI) update (see Appendix A.5 for a formal proof).

This equivalence is significant: Particle-based methods typically struggle in higher-dimensional settings, as we will observe in Section 5.1, while OVI performs competitively with ES on higher-dimensional Brax tasks (Figure 6). On the other hand, both OVI and ES, being parametric methods with Gaussian distributions, are limited to finding a single optimum, whereas cCBO can discover multiple optima. We

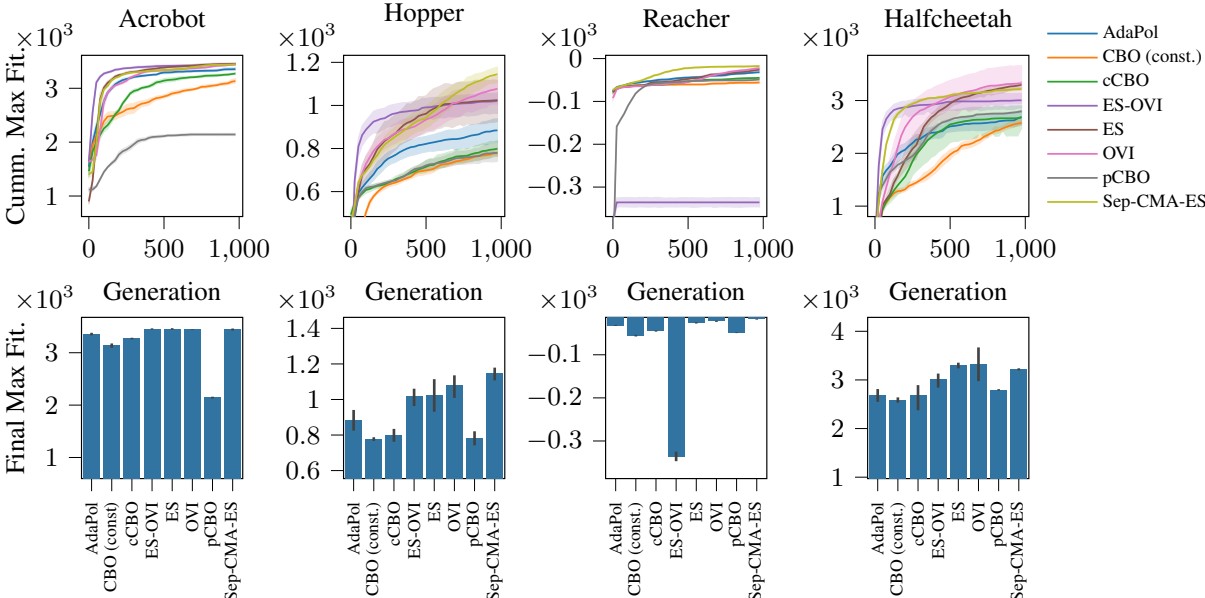

*Figure 4.* **AdaPol improves upon CBO in higher dimensional tasks. OVI is competitive with CMA-ES.** Evaluation on Brax tasks with 10 seeds each. The shaded areas show 95% CIs across 10 seeds, hyper-parameters were optimized for each method per each task.

thus combine both approaches to obtain a method capable of finding multiple optima in higher-dimensional problems.

**AdaPol & SchedPol**   Since CH is equivalent to OVI, we can interpolate between the two regimes by varying $\lambda$: We propose starting with $\lambda = 1$ (CH/OVI behavior) to quickly reach a promising region, and then reduce $\lambda$ to enable multi-modal exploration via cCBO. A fixed schedule for $\lambda$ is an option, but determining when to transition is problem-dependent. We thus propose an adaptive scheme inspired by JADE (Zhang & Sanderson, 2007). At each iteration, we allocate particles to two strategies, CH ($\lambda = 1$) and cCBO ($\lambda < 1$), in proportion to the number of successful particles each strategy produced over the previous $N_G$ generations, where success is defined as having fitness in the top $p\%$ of the current population. If either strategy's success rate drops to zero, we allocate a small fraction of particles to it for continued exploration. We refer to this adaptive method as AdaPol and to the fixed-schedule variant as SchedPol.

## 5. Experiments

We first evaluate all methods on the common BBOB and Brax tasks, then investigate the behavior of ES-OVI specifically on the Brax tasks, and finally propose the application of multi-modal optimizers to LLM merging. Additional experimental details are shown in Appendix B, along with runtime comparisons.[1]

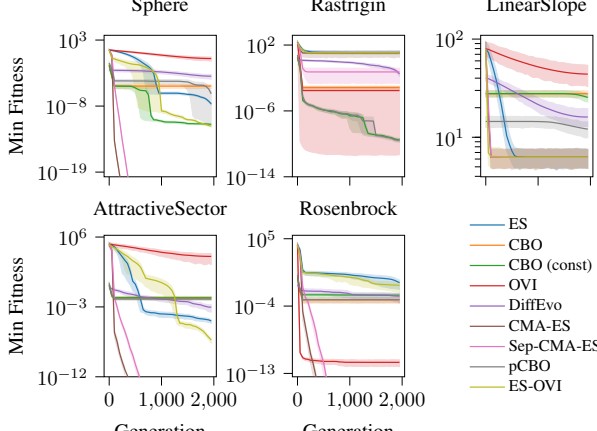

*Figure 5.* **Hybrid optimizers can outperform base methods.** Evaluation on a selection of 2D BBO tasks. Note that in multiple problems, ES-OVI performs better than either ES or OVI. The shaded areas show 95% CIs across 10 random seeds, hyper-parameters were optimized for each method in each problem.

### 5.1. Benchmarks

To our knowledge, neither CBO nor OVI has been evaluated on the popular BBO Benchmark (BBOB) (Hansen et al., 2009) or Brax (Freeman et al., 2021) benchmark. We thus evaluate both the existing and newly proposed methods in these two settings. We implement them in JAX (Bradbury et al., 2018), using the evosax library (Lange, 2022), reusing the existing implementations when available. We also evaluate a variant of CBO that uses a constant diffusion term $\sigma \epsilon_t^i$ instead of $\sigma \|x_t^i - m_t\| \epsilon_t^i$, denoted as "CBO (const.)".

---

[1]Our implementation is available at https://github.com/JohannesAck/bridging_optimizers.

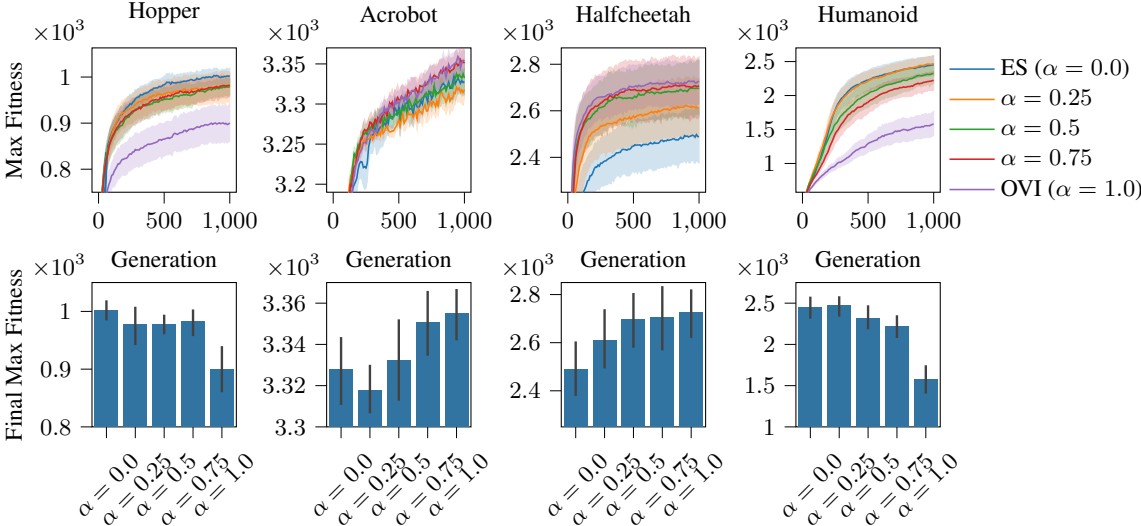

*Figure 6.* **ES-OVI allows us to tune the optimization behavior for each environment.** Performance on Brax of ES-OVI with different interpolation-coefficients $\alpha$. $\alpha = 1.0$ corresponds to OVI, $\alpha = 0.0$ corresponds to ES. 20 trials, 90% CIs.

**Black-Box Optimization Benchmark** For low-dimensional problems, we evaluate each method on 23 of the BBOB benchmark problems (Hansen et al., 2009). The BBOB benchmark consists of a series of parametric problems, different in each initialization. As we use Brax for higher-dimensional evaluation, we here first use the 2D variants of the problems. For each problem, we optimize the hyperparameters of each method by grid search with 10 initializations each and select the best configuration based on mean fitness after 1000 generations. A subset of the problems is shown in Figure 5. In Appendix B.2, we show the results for the remaining tasks, along with the hyperparameter combinations considered for each method. We highlight some findings: In multimodal tasks, such as "RastriginOriginal", the particle-based CBO and pCBO perform particularly well, as expected for particle-based methods. There are cases in which both OVI and ES fail, but ES-OVI performs well, such as "Attractive Sector". Conversely, there are cases where it performs worse than either, such as "GriewankRosebrock". This shows the utility of having another "knob" to control the convergence behavior, even when hyperparameters are optimized. We also repeated the experiments for higher dimensionalities and show the results in Appendix C.

**Brax** To evaluate the methods in a higher-dimensional setting, we use the Brax (Freeman et al., 2021) implementations of four classical control tasks. In these tasks, the goal is to find parameters $x$ for a locomotion policy represented by a two-layer MLP with 32 units per layer. The problem dimensionality is thus approximately $d \approx 1000$, with the exact dimensionality varying due to different input and output dimensionalities of each robot. The input of the

MLP is the current state of the robot $s$ and the output is the action $a = f(x)(s)$ representing the torque applied to each actuator of the robot. We want to maximize the sum of the reward collected over an episode. The results are shown in Figure 4. We find that AdaPol tends to perform better than other CBO variants. Additionally, ES-OVI ($\alpha = 0.5$) initially provides better performance than other methods, but tends to achieve a worse final result.

### 5.2. ES-OVI

To evaluate ES-OVI further, we perform experiments with different interpolation values $\alpha$ on the Brax tasks. The results are shown in Figure 6. While both ES ($\alpha = 0.0$) and OVI ($\alpha = 1.0$) have clear failure modes on different tasks, we find that setting an intermediate $\alpha = 0.75$ performs well across tasks. We have also seen that on the Rosenbrock function, ES converges to a flatter region while OVI converges to the sharp minimum, with ES-OVI interpolating between them. To evaluate whether this behavior also occurs in higher dimensions, we perform experiments on the Acrobot task. After convergence, we evaluate the objective under Gaussian disturbances of the found parameters $x$, defined as $\mathbb{E}[F(x + \epsilon)]$ with $\epsilon \sim \mathcal{N}(0, \sigma^2 I)$. The results are shown in Figure 7. OVI indeed finds a sharper, more sensitive solution than ES, while intermediate $\alpha$ values interpolate between them. As parameter sensitivity in reinforcement learning is connected to robustness to action disturbances or changed dynamics (Lee & Yoon, 2025), we also evaluate the performance under observation noise $a = f(x)(s + \epsilon)$ and action noise $a = f(x)(s) + \epsilon$. The results, shown in Figure 7, indeed show a higher sensitivity to action noise of OVI than ES. Interestingly, ES-OVI achieves a better

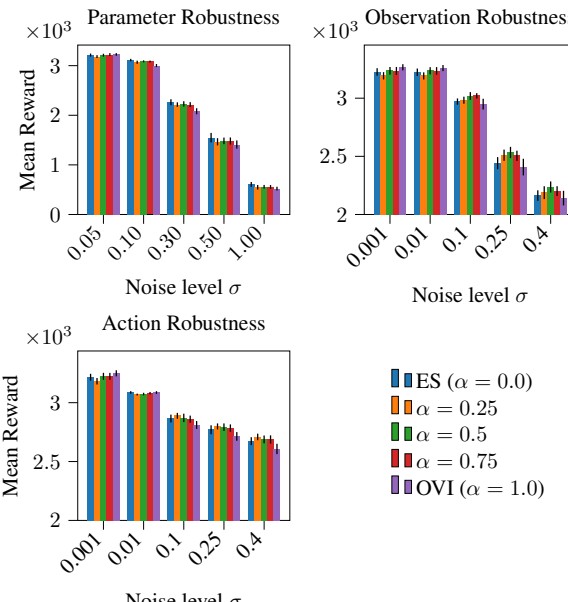

*Figure 7.* **ES-OVI allows us to trade performance vs robustness.** Robustness on the Acrobot task, trained with ES-OVI with different $\alpha$ values. Mean across 20 different random seeds with bootstrapped 95% CIs.

robustness than either OVI or ES under strong action or observation disturbances. This experiment also gives us some guidance on how we can approach picking the hyperparameter $\alpha$: When we have less confidence in the fidelity of our function $F$, we may use a smaller $\alpha$ to obtain a more robust solution. When $F$ is reliable, we can use a larger $\alpha$ to obtain a better performance.

### 5.3. Evolutionary model merging

To evaluate the utility of the combined parametric and particle-based AdaPol and SchedPol, we propose to use model merging with limited evaluations as a benchmark. Model merging (Wortsman et al., 2022; Ilharco et al., 2023) is a post-training method for LLMs, which combines multiple fine-tuned (FT) versions of the same model with different desirable characteristics. It interpolates the parameters $\phi$ with weighting $x_i$, $\phi_{\text{Merge}} = \phi_{\text{Base}} + \sum_i x_i(\phi_{\text{FT,i}} - \phi_{\text{base}})$. These weights may be chosen by layer, resulting in a high-dimensional problem, making the manual choice of $x$ difficult. Evolutionary Model Merge (Akiba et al., 2025) thus uses CMA-ES to obtain a weighting. Their objective function evaluates the model on 1096 machine-translated Japanese samples from GSM8K, making each evaluation costly. Instead, we propose to evaluate each candidate only on a small, fixed subset of problems, significantly reducing the cost per candidate. Unfortunately, this turns the model merging problem into a multi-modal problem setting: Some optima of the objective overfit to the specific problems in the subset, while others generalize better. By obtaining multiple

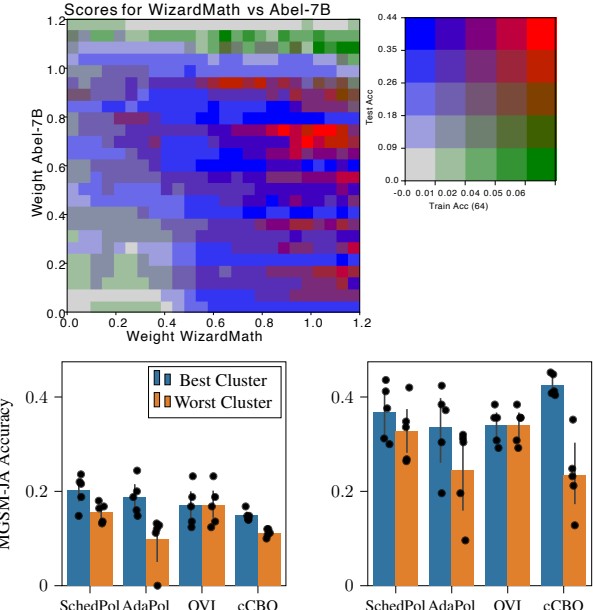

*Figure 8.* **Multimodal optimizers are advantageous in model merging with limited data.** Top: Test and train accuracy for different merges, showing that model merging is a multi-modal problem. Bottom: Test accuracy when initializing centered on 0.0 (left) or with a broader init at 0.65 (right). Both show the mean across five random seeds, with bootstrapped 95% CIs. Multimodal cCBO is advantageous for the better init, our proposed SchedPol is competitive in both.

optima on the small subset and evaluating each on the full training dataset only once at the end, we can significantly reduce the computational cost.

We combine Mistral 7B (Jiang et al., 2023) with three fine-tuned versions, resulting in a 33-dimensional problem. Testing is done on 250 hand-translated MGSM questions (Shi et al., 2023), while we train on a fixed subset of 64 questions from the machine-translated GSM8K. To first show that model merging with a limited dataset is indeed a multi-modal problem, we visualize the training and test accuracies for a 2D slice of the parameter space in Figure 8 (top). Next, we evaluate OVI, cCBO, AdaPol and SchedPol with a tuned $\lambda$ schedule, each with two different initial distributions. In Figure 8 (bottom left), we show the test accuracy for two different initializations. The results show: When provided with a good initialization, treating model merging with limited data as a multi-modal problem and using cCBO performs significantly better than the unimodal OVI. Conversely, when initialized in a bad region of parameter space, cCBO performs worse than OVI. By combining both methods, either with a manually tuned SchedPol or AdaPol, we can perform competitively in both cases, as the OVI behavior first reaches a good region of parameter space in which cCBO can identify multiple optima. All methods are unable to match the accuracy of the method proposed by Akiba et al. (2025) due to overfitting the training dataset.

## 6. Related Work and Limitations

While we are not aware of any papers investigating the connection of ES, OVI, and CBO, other connections between optimizers have previously been investigated. Braun et al. (2025) investigated the combination of Stein Variational Gradient Descent (SVGD) with ES, replacing the gradient in SVGD with an estimate provided by ES or CMA-ES. Xu et al. (2019) proposed a combination of particle swarm optimization (PSO) and CMA-ES. However, PSO does not allow for any theoretical analysis of its convergence and the combination, while effective, appears heuristic. We thus focus on CBO and its variants instead, due to their advantageous theoretical properties and their relations to ES and OVI. Learned Evolution Strategy (LES) (Lange et al., 2023) investigated how the parameter update in ES methods can be meta-learned. In principle, LES could learn both the update rule of ES as well as OVI, and their combinations in our proposed ES-OVI method. However, by uncovering these connections manually we can explicitly control their properties. LES does not extend to particle based methods, but it would be interesting to consider similar meta-learning methods applied to the interaction matrix in our setup. Carrillo et al. (2021) introduced multiple changes for CBO to allow application to high-dimensional problem settings. However, it does not permit us to obtain multiple optima in high-dimensional problems, which aim for by combining cCBO and OVI. Information Geometric Optimization (IGO) (Ollivier et al., 2017) provides a unified framework for multiple black box optimizers, including NES, CMA-ES, the cross-entropy method, and related parametric methods. To our understanding, neither OVI nor the non-parametric CBO or DiffEvo, which we consider, fit into the IGO framework.

MERGE[3] (Mencattini et al., 2025) proposes to improve the sample efficiency of Evolutionary Model Merging by using a reduced dataset for training, similar to our experiments on evolutionary model merging. Orthogonally to our work, they focus on using item response theory to more accurately predict the performance on the whole dataset from evaluations on the limited dataset, while we instead change the optimizer to allow us to obtain multiple optima.

### 6.1. Limitations

A key limitation of our work is that we consider only optimizers based on spherical Gaussian distributions and thus cannot cover methods which adapt the covariance matrix during training, such as CMA-ES (Hansen, 2016). Our hybrid methods also introduce new hyper-parameters, $N_\mathrm{G}$ in the case of AdaPol and $\alpha$ in the case of ES-OVI. While we found $N_\mathrm{G}$ to not be very sensitive, $\alpha$ heavily affects the convergence characteristics of ES-OVI, as discussed in Section 5.2. We believe that addressing these limitations would be interesting avenue for future work.

## 7. Conclusion

We investigated connections between multiple parametric and non-parametric Black-Box Optimizer. We proposed a unified view, revealing that the considered BBO methods differ primarily in two design choices: aggregation (expectation vs. log-sum-exp, controlling sharpness preference) and consensus scope (global vs. local, controlling modality). By interpolating along these axes, we obtained new methods that can outperform either of their endpoints. We hope that these perspectives will help guide both method selection for practitioners and future algorithm design.

## Impact Statement

Our work focuses on black-box optimizers. Optimizers are general purpose tools with many different applications, both positive and negative ones. There are thus many potential societal consequences, none of which we feel must be specifically highlighted here.

## Acknowledgements

We would like to thank Robert Lange, Stefania Druga, and Maxence Faldor for helpful advice and discussions.

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

# A. Appendix

## A.1. Fitness Shaping

Rather than using raw fitness values directly, implementations of ES and other BBO methods typically apply a fitness shaping transformation before computing updates. A common choice is rank-based shaping (Wierstra et al., 2014; Salimans et al., 2017), which replaces fitness values with a function of their ranks:

$$\mathcal{T}_{\mathrm{R}}(x^i) = \frac{\mathrm{rank}(x^i)}{N} - 0.5, \tag{7}$$

where $\mathrm{rank}(x^i) \in \{1, \ldots, N\}$ denotes the rank of sample $x^i$ among the $N$ samples in the current generation ($\mathrm{rank}(x^i) = j$ if $F(x^i)$ is the $j$-th smallest fitness value). Using (7) makes the optimization invariant to any monotonic transformation of the fitness function. An alternative is standardization, which centers and normalizes the fitness values:

$$\mathcal{T}_{\mathrm{Z}}(x^i) = \frac{F(x^i) - \hat{\mu}_F}{\hat{\sigma}_F}.$$

where $\hat{\mu}_F$ and $\hat{\sigma}_F$ are the sample mean and standard deviation of the fitness values $\{F(x^i)\}_{i=1}^N$. Both transformations yield updates with controlled magnitude, reducing sensitivity to the scale of the objective and facilitating learning rate selection (Wierstra et al., 2008).

## A.2. Stochastic Smoothing

Given a smoothing distribution with full support on $\mathbb{R}^d$ and differentiable density $\pi$, the smoothed objective is $J_\pi^{\mathrm{SS}}(\theta) = \mathbb{E}_{\epsilon \sim \pi}[F(\theta + \epsilon)]$. Using the likelihood-ratio estimator (Rubinstein, 1986; Glynn, 1990; Glasserman, 2004), the gradient with respect to $\theta$ is

$$\nabla_\theta J_\pi^{\mathrm{SS}}(\theta) = \mathbb{E}_{\epsilon \sim \pi}\left[F(\theta + \epsilon)\nabla_\epsilon - \log \pi(\epsilon)\right].$$

Crucially, $J_\pi^{\mathrm{SS}}(\theta)$ is differentiable in $\theta$ even when $F$ is not. For a Gaussian smoothing distribution $\pi = \mathcal{N}(0, \sigma^2 I)$, the score simplifies to $\nabla_\epsilon - \log \pi(\epsilon) = \epsilon/\sigma^2$, yielding

$$\nabla_\theta J^{\mathrm{SS}}(\theta) = \frac{1}{\sigma}\mathbb{E}_{\epsilon \sim \mathcal{N}(0,I)}\left[F(\theta + \sigma\epsilon)\epsilon\right].$$

The gradient descent updates $\theta_{t+1} = \theta_t - \eta\nabla_\theta J^{\mathrm{SS}}(\theta_t)$ are implemented via the Monte Carlo estimator in (SS).

## A.3. Optimization Via Integration

The OVI updates can be derived from a Bayesian perspective. A parametric family of reference distributions $\Pi$ is chosen. At each iteration, the current search distribution $\pi_t \in \Pi$ is tilted toward regions of low loss via $\tilde{\pi}_{t+1}(x) \propto \exp(-\beta F(x))\pi_t(x)$, then projected back onto $\Pi$ by minimizing the KL divergence $\mathsf{KL}(\tilde{\pi}_{t+1}\|\pi)$ over $\pi \in \Pi$. If $\Pi = \{\mathcal{N}(\theta, \sigma^2 I)\}_{\theta \in \mathbb{R}^d}$ with fixed $\sigma^2$, the KL projection reduces to computing the mean of the tilted distribution:

$$\theta_{t+1} = \mathbb{E}_{\tilde{\pi}_{t+1}}[x] = \frac{\mathbb{E}_{\pi_t}[xe^{-\beta F(x)}]}{\mathbb{E}_{\pi_t}[e^{-\beta F(x)}]}.$$

Employing the self-normalized importance sampling MC estimator yields (OVI).

The OVI updates also correspond to gradient descent on $J^{\mathrm{OVI}}$ with step size $\sigma^2$. To see this, rewrite (OVI) as $\theta_{t+1} = \theta_t + \sum_{i=1}^N w^i \epsilon^i$, where $w^i = e^{-\beta F(x^i)}/\sum_j e^{-\beta F(x^j)}$ and $\epsilon^i = x^i - \theta_t$. Note that $J^{\mathrm{OVI}}(\theta) = -\log Z(\theta)$ where $Z(\theta) = \mathbb{E}_\epsilon[e^{-\beta F(\theta+\epsilon)}]$, $\epsilon \sim \mathcal{N}(0, \sigma^2 I)$. Differentiating, $\nabla_\theta J^{\mathrm{OVI}}(\theta) = -\nabla_\theta Z(\theta)/Z(\theta)$, where

$$\nabla_\theta Z(\theta) = \frac{1}{\sigma^2}\mathbb{E}_\epsilon\left[\epsilon e^{-\beta F(\theta+\epsilon)}\right],$$

and thus $\nabla_\theta J^{\mathrm{OVI}}(\theta) = -\frac{1}{\sigma^2}\sum_i w^i \epsilon^i$ and $\theta_{t+1} = \theta_t - \sigma^2\nabla_\theta J^{\mathrm{OVI}}(\theta_t)$.

### A.4. Denoising Diffusion Models and Diffusion Evolution

In diffusion models, one defines a forward noising process $x_t = \sqrt{\alpha_t}x_0 + \sqrt{1-\alpha_t}\epsilon$ with $\epsilon \sim \mathcal{N}(0, I)$ and $\alpha_t \in (0, 1]$ decreasing over time. The conditional distribution of $x_t$ given $x_0$ is $q_{\alpha_t}(x_t \mid x_0) = \mathcal{N}(x_t; \sqrt{\alpha_t}x_0, (1-\alpha_t)I)$. Given a *fixed* target distribution $\pi(x_0) \propto \exp(-\beta F(x_0))$ over clean samples, the optimal denoiser minimizing $\mathbb{E}[\|\hat{x}_0(x_t) - x_0\|^2]$ is the conditional expectation

$$m_t(x_t) = \mathbb{E}_\pi[x_0 \mid x_t] = \frac{\int x_0 \pi(x_0) q_{\alpha_t}(x_t \mid x_0)\mathrm{d}x_0}{\int \pi(x_0) q_{\alpha_t}(x_t \mid x_0)\mathrm{d}x_0}. \tag{8}$$

Instead of computing this integral exactly, Diffusion Evolution approximates it using the current particle population via (5), where the kernel $k_t(x, y) \propto q_{\alpha_t}(x \mid y)$.

### A.5. Equivalence of CH and OVI

We show that CH and OVI are the same particle method.

Given particles $\{x_t^i\}_{i=1}^N$, CH defines the (global) consensus

$$m_t = \frac{\sum_{i=1}^N x_t^i \exp(-\beta F(x_t^i))}{\sum_{i=1}^N \exp(-\beta F(x_t^i))}.$$

CH then resamples

$$x_{t+1}^i = m_t + \sigma \epsilon_{t+1}^i, \qquad \epsilon_{t+1}^i \overset{\text{i.i.d.}}{\sim} \mathcal{N}(0, I).$$

OVI samples $x_t^i = \theta_t + \sigma \epsilon_t^i$ and updates the mean by the same weighted average:

$$\theta_{t+1} = \frac{\sum_{i=1}^N x_t^i \exp(-\beta F(x_t^i))}{\sum_{i=1}^N \exp(-\beta F(x_t^i))} = m_t,$$

then resamples $x_{t+1}^i = \theta_{t+1} + \sigma \epsilon_{t+1}^i$.

Assume both methods start from the same $\theta_0$ and use the same Gaussian perturbations $\{\epsilon_t^i\}$. Then at $t = 0$ they generate the same particles $\{x_0^i\}$. If $\{x_t^i\}$ coincide at some iteration $t$, then both compute the same $m_t$, and with the same perturbations they generate the same $\{x_{t+1}^i\}$. By induction, CH and OVI produce identical particle trajectories for all $t$, and their centers satisfy $\theta_{t+1} = m_t$.

### A.6. ES as (MU) update

We show how ES can be formulated as a particle update of the form (MU), provided that the sampling perturbations have zero empirical mean.

Let $\theta_t \in \mathbb{R}^d$ be the search distribution mean and $\sigma > 0$ fixed. Sample perturbations $\epsilon_t^i$ and candidates $x_t^i = \theta_t + \sigma \epsilon_t^i$. Assume that the perturbations satisfy $\sum_{i=1}^N \epsilon_t^i = 0$, e.g. antithetic sampling with even $N$, for each $t$. The (mean-only) ES update corresponding to (NES) is

$$\theta_{t+1} = \theta_t - \frac{\eta}{N\sigma} \sum_{i=1}^N g_t^i \epsilon_t^i, \tag{9}$$

where $g_t^i = F(x_t^i)$. Under $\sum_{i=1}^N \epsilon_t^i = 0$, subtracting any constant baseline from the scores leaves the ES update unchanged, so (9) is equivalent to the baseline-centered form

$$\theta_{t+1} = \theta_t - \frac{\eta}{N\sigma} \sum_{i=1}^N (g_t^i - \bar{g}_t)\epsilon_t^i, \qquad \bar{g}_t = \frac{1}{N} \sum_{i=1}^N g_t^i.$$

Note that $\theta_t$ can be recovered from $\{x_t^i\}$: $\frac{1}{N}\sum_{i=1}^N x_t^i = \theta_t$.

Define per-particle weights

$$w_t^i = \frac{1}{N} - \frac{\eta}{N\sigma^2}(g_t^i - \bar{g}_t), \tag{10}$$

which satisfy $\sum_{i=1}^{N} w_t^i = 1$. For the consensus $m_t = \sum_{i=1}^{N} w_t^i x_t^i$, using $x_t^i = \theta_t + \sigma \epsilon_t^i$ and $\sum_i \epsilon_t^i = 0$, we obtain

$$m_t = \theta_t + \sigma \sum_{i=1}^{N} w_t^i \epsilon_t^i = \theta_t - \frac{\eta}{N\sigma} \sum_{i=1}^{N} (g_t^i - \bar{g}_t)\epsilon_t^i = \theta_{t+1}.$$

Therefore ES can be written as a particle-only update: (i) compute $m_t$ from particles using weights (10), (ii) resample $x_{t+1}^i = m_t + \sigma \epsilon_{t+1}^i$. This matches (MU) with $K_t^{i,j} = 1$, $\mu_t = 0$, $\lambda_t = 1$, and $s(\Delta) = 1$.

If $\epsilon_t^i \overset{\text{i.i.d.}}{\sim} \mathcal{N}(0, I)$ without enforcing $\sum_i \epsilon_t^i = 0$, then the particle mean is

$$\hat{\theta}_t = \frac{1}{N} \sum_{i=1}^{N} x_t^i = \theta_t + \sigma \bar{\epsilon}_t, \quad \bar{\epsilon}_t = \frac{1}{N} \sum_{i=1}^{N} \epsilon_t^i \sim \mathcal{N}\left(0, \frac{1}{N}I\right).$$

If one replaces $\theta_t$ by $\hat{\theta}_t$, as in the particle ES update from (MU), the resulting iterates differ from the true ES updates from (9) by terms proportional to $\bar{\epsilon}_t$. So the particle representation for ES remains a good approximation even without antithetic sampling, as long as $N \gg 1$.

### A.7. cCBO as (MU) update

cCBO (Bungert et al., 2025) maintains soft assignments $p_t^{i,c} \in [0,1]$ of particle $i$ to cluster $c \in \{1, \dots, N_C\}$, with $\sum_{c=1}^{N_C} p_t^{i,c} = 1$, and defines cluster centers as fitness-weighted averages of the particles. We show that its per-particle consensus can be written in the MU form (MU) by choosing a cluster-induced low-rank interaction matrix $K_t$.

Let $\Psi(F) = \exp\{-\beta F\}$ and define the cluster normalizers

$$Z_t^c := \sum_{j=1}^{N} p_t^{j,c} \Psi(F(x_t^j)).$$

Define the cluster-induced interaction (for indices $i, j$)

$$K_t^{i,j} := \sum_{c=1}^{N_C} \frac{p_t^{i,c} p_t^{j,c}}{Z_t^c}. \tag{11}$$

Then the MU unnormalized weights satisfy

$$a_t^{i,j} = \Psi(F(x_t^j)) K_t^{i,j} = \sum_{c=1}^{N_C} p_t^{i,c} \frac{p_t^{j,c} \Psi(F(x_t^j))}{Z_t^c},$$

where

$$\sum_{j=1}^{N} a_t^{i,j} = \sum_{c=1}^{N_C} p_t^{i,c} = 1,$$

hence $w_t^{i,j} = a_t^{i,j}$. The corresponding MU consensus becomes

$$m_t^i = \sum_{j=1}^{N} w_t^{i,j} x_t^j = \sum_{c=1}^{N_C} p_t^{i,c} \underbrace{\left(\sum_{j=1}^{N} x_t^j \frac{p_t^{j,c} \Psi(F(x_t^j))}{Z_t^c}\right)}_{=: \, c_t^c},$$

which equals the cCBO consensus $m_{c,t}^i = \sum_{c=1}^{N_C} p_t^{i,c} c_t^c$, where each center $c_t^c$ is a fitness-weighted average over the particles assigned to cluster $c$.

cCBO additionally specifies how to update the assignments $p_t^{i,c}$ over time; in our framework this corresponds to specifying the time evolution of the interaction matrix $K_t$.

*Table 2.* Hyperparameters used in BBOB tasks. Other hyper-parameter were optimized by grid-search over values shown below.

| Hyperparameter | Value |
|---|---|
| Generations | 2000 |
| Population | 256 |
| DE Mapping | Energy |
| DE $\alpha$ schedule | cosine |
| ES-OVI $\alpha$ | 0.5 |

## B. Experiment Details

### B.1. Implementation Details

We implement all methods in Jax (Bradbury et al., 2018) based on the existing methods in evosax (Lange, 2022). Due to Jax's just-in-time compilation, this allows the evaluation of multiple candidates $x$ and even multiple algorithm hyper-parameters in parallel, significantly decreasing the computational cost of hyper-parameter optimization. Unless stated otherwise below, we use the default hyper-parameters provided by evosax. For pCBO and cCBO we use Gaussian kernels $k(x, y) = \exp\left(-\frac{\|x-y\|_2^2}{2\kappa^2}\right)$, with the squared Euclidean norm $\|\cdot\|_2^2$.

**AdaPol, SchedPol**   In AdaPol we follow JADE (Zhang & Sanderson, 2007) and allocate particles to each of two strategy ($\lambda = 1$ and $\lambda = 0.1$) proportionally to the success rate of each over the last $N_G$ generations. The success rate is defined as the percentage of candidates belonging to each strategy in the top $p\%$ of the total particles. To reallocate particles to strategies as needed we change as few particles as possible, and when changing from OVI behavior to cCBO behavior randomly initialize the cluster assignments $p^{i,j}$. When all particles are assigned to OVI, we randomly assign 1/3 of particles to cCBO for exploration. In this case, we randomly sample new cluster centers $c^j$ uniformly from the OVI particles reassigned to OVI and randomly initialize all cluster assignments $p^{i,j}$, by sampling uniform $p_{i,j} \sim \mathcal{U}(0, 1)$ and then normalizing for each particle. When using SchedPol this only happens once when changing from $\lambda = 1$ to $\lambda < 1$. We also use this initialization in the first step of our cCBO implementation.

### B.2. BBOB

We show the full results on all tested BBOB functions in Figure 9.

#### B.2.1. HYPERPARAMETER OPTIMIZATION

Fixed hyper-parameters are shown in Table 2. We tune hyper-parmeters for each problem with 10 initializations and grid-search over the following parameters.

- ES: $\sigma \in \{0.0001, 0.0003, 0.001, 0.003, 0.01\}$, $lr \in \{0.00001, 0.00003, 0.0001, 0.0003, 0.001, 0.003\}$

- CMA-ES: $\sigma \in \{0.0001, 0.0003, 0.001, 0.003, 0.01\}$, $c_M \in \{0.8, 0.9, 1.0, 1.1, 1.2\}$

- Sep-CMA-ES: $\sigma \in \{0.0001, 0.0003, 0.001, 0.003, 0.01\}$, $c_M \in \{0.8, 0.9, 1.0, 1.1, 1.2\}$

- Diffusion Evolution $\sigma \in \{0.0001, 0.0003, 0.001, 0.003, 0.01\}$, fitnessmap temperature $\in \{0.6, 0.7, 0.8, 0.9, 1.0\}$

- CBO $\sigma \in \{0.0001, 0.0003, 0.001, 0.003, 0.01\}$, $\lambda \in \{0.001, 0.1, 0.2, 0.3, 0.5\}$

- pCBO $\sigma \in \{0.0001, 0.001, 0.01\}$, $\lambda \in \{0.1, 0.2, 0.001, 0.3, 0.5\}$, $\kappa \in \{1, 2, 5\}$

- OVI $\sigma \in \{0.0001, 0.0003, 0.001, 0.003, 0.01\}$, $\beta \in \{0.5, 0.75, 1.0, 1.25, 1.5\}$

- ES-OVI $\sigma \in \{0.0001, 0.0003, 0.001, 0.003, 0.01\}$, $lr \in \{0.00001, 0.00003, 0.0001, 0.0003, 0.001, 0.003\}$

- AdaPol $\sigma \in \{0.001, 0.003, 0.01, 0.03, 0.1, 0.3\}$

In addition, for all methods we also evaluate using the raw scores vs rank-based fitness transformation, and constant $\sigma$ vs decayed $\sigma$. We choose the best hyper-parameters by best mean fitness after 1000 generations.

*Table 3.* Hyperparameters used in Brax tasks. Other hyper-parameter were optimized by grid-search over values shown below.

| Hyperparameter | Value |
| --- | --- |
| Generations | 1000 |
| Episode Length | 500 |
| Population | 256 |
| ES-OVI $\alpha$ | 0.5 |
| pCBO, cCBO $\lambda$ | 0.5 |
| cCBO $N_{\mathrm{C}}$ | 4 |

## B.3. Brax experiments

We run each environment for 500 steps per episode, diverging from the commonly used 1000 steps per episode to reduce computational cost. We were unable to obtain a satisfying result using Diffusion Evolution on the Brax tasks and thus excluded it from the results. For all CBO based methods we here use the constant noise term as in CBO (const.), as we found it to be beneficial to prevent premature convergence to suboptimal policies.

### B.3.1. HYPERPARAMETER OPTIMIZATION

We optimize the hyper-parameters for each approach in each task by grid-search, other hyper-parameters are shown in Table 3. For the Brax tasks, we use 10 seeds each for Acrobot, Hopper and Reacher and 5 seeds each for Half-Cheetah. We choose the best hyper-parameter by max-fitness after 1000 generations. We perform grid-search over the following hyperparameters:

- ES: $\sigma \in \{0.01, 0.03, 0.1, 0.3\}$, $lr \in \{0.0001, 0.0003, 0.001, 0.003, 0.01, 0.03, 0.1\}$

- Sep-CMA-ES: $\sigma \in \{0.1, 0.3, 1.0\}$

- CBO $\sigma \in \{0.01, 0.03, 0.1, 0.3\}$, $\lambda \in \{0.1, 0.2, 0.001, 0.3, 0.5\}$

- pCBO $\sigma \in \{0.01, 0.03, 0.1, 0.3\}$, $\kappa \in \{0.1, 1, 10, 100\}$

- cCBO $\sigma \in \{0.01, 0.03, 0.1, 0.3\}$, $\kappa \in \{0.1, 1, 10, 100\}$

- OVI $\sigma \in \{0.001, 0.003, 0.01, 0.03, 0.1, 0.3, 1.0\}$

- ES-OVI $\sigma \in \{0.01, 0.03, 0.1, 0.3, 1.0\}$, $lr \in \{0.00005, 0.0001, 0.0003, 0.001, 0.003, 0.01\}$

- AdaPol $\sigma \in \{0.001, 0.003, 0.01, 0.03, 0.1, 0.3\}$

For ES we also tried to use a linear decay for the sampling distributions $\sigma$, as well as the learning rate, but we did not find a benefit of doing so and thus present the results with constant $\sigma$ and learning rate. For AdaPol we use $N_G = 100$, $\lambda = 0.3$. For OVI we use $J^{\mathrm{OVIZ}}$ in the Brax experiments, i.e. $\beta = 1/\hat{\sigma_F}$ in each iteration. With a fixed $\beta$ we were unable to obtain useful policies.

## B.4. Model Merging Details

Following (Akiba et al., 2025), we implement model merging based on the DARE-TIES method and use the same base model *Mistral-7B-v0.1* (Jiang et al., 2023) and three finetunes: *WizardMath-7B-V1.1* (Luo et al., 2025), *shisa-gamma-7b-v1* (augmxnt, 2023), and *Abel-7B-002* (Chern et al., 2023). By implementing both the merging and evaluation on multiple GPUs, we can reduce the costly CPU-GPU transfers needed by other methods. However, this requires holding four 7B models in GPU memory, requiring 56GB for just the bf16 weights. To reduce this memory burden but avoid CPU-GPU transfers, we shard the base and finetuned models across 4GPUs and all-reduce the result across GPUs. We then perform the evaluation using a modified, stateful version of lmeval (Gao et al., 2024). For the black-box optimizers we reuse our Jax implementation based on evosax (Lange, 2022). Hyperparameters are shown in Table 4. We do not decay the noise during training as we found it not to be beneficial. Further, while we evaluated the combination of DARE (Yu et al., 2024) and

*Table 4.* Hyperparameters used for model merging experiments

| Hyper-parameter | Values |
|---|---|
| Train Dataset Size | 64 |
| Layer Reduction Factor | 4 |
| Population Size | 128 |
| Init range | [-0.1, 0.1] *or* [0.5,0.8] |
| Weight range | [-0.2,2.0] |
| $\sigma$-init | 0.4 |
| Generations | 60 |
| Cluster $N_C$ | 4 |
| cCBO $\kappa$ | 6 |
| cCBO $\alpha$ | 4 |
| cCBO $\lambda$ | 0.3 |
| SchedPol OVI generations | 40 |

*Table 5.* Runtime of different methods for 1000 iterations with population size 256, averaged across 10 trials each.

| | OpenAI-ES | OVI | ESOVI | CMA-ES | Sep-CMA-ES | CBO | DiffEvo | AdaPol |
|---|---|---|---|---|---|---|---|---|
| D=5 | 4.0s | 4.1s | 5.8s | 9.9s | 7.7s | 5.9s | 4.9s | 13.8s |
| D=50 | 4.0s | 4.1s | 5.8s | 10.3s | 7.7s | 6.0s | 5.0s | 13.8s |
| D=1000 | 4.1s | 4.2s | 6.1s | 14.5s | 7.8s | 6.2s | 5.1s | 13.8s |
| D=5000 | 4.2s | 4.3s | 6.1s | 149s | 8.0s | 6.5s | 5.7s | 14.0s |

TIES (Yadav et al., 2023), we found in our experiments that the dropout introduced by DARE was not advantageous and thus only use TIES. Further, we do not use the top $\Delta$ trimming used in TIES and only retain the sign consensus mechanism, as we did not find it to be beneficial in our setting.

### B.5. Runtime Comparison

To investigate the impact of our proposed methods on the runtime of the algorithm, we run each method for 1000 iterations with a random fitness function and population size 256. We use a just-in-time compiled implementation in JAX and perform our benchmarks on an NVIDIA RTX 2080 Super GPU. The results are shown in Table 5. ESOVI takes significantly longer than either OpenAI-ES or OVI, and AdaPol takes significantly longer than OVI or CBO. However, in black-box optimization the evaluation of the function $F$ we are optimizing is typically significantly more expensive than the black-box optimizer. For example, in our model merging experiments, evaluating a single candidate $x$ takes 12 seconds on an NVIDIA H100 GPU, while the time taken to generate the new candidate population with 128 candidates is less than a second.

# C. BBOB results for different dimensionalities

To investigate the performance of the different optimizers as we increase the problem dimensionality, we repeat the BBOB experiments on higher dimensional variants of the benchmark problems. We perform the same hyper-parameter optimization separately for each dimensionality, as discussed in Section B.2.1. As shown in Figures 10, 11, 12, 13, 14, we largely observe a similar trend for higher dimensional variants of the tasks.

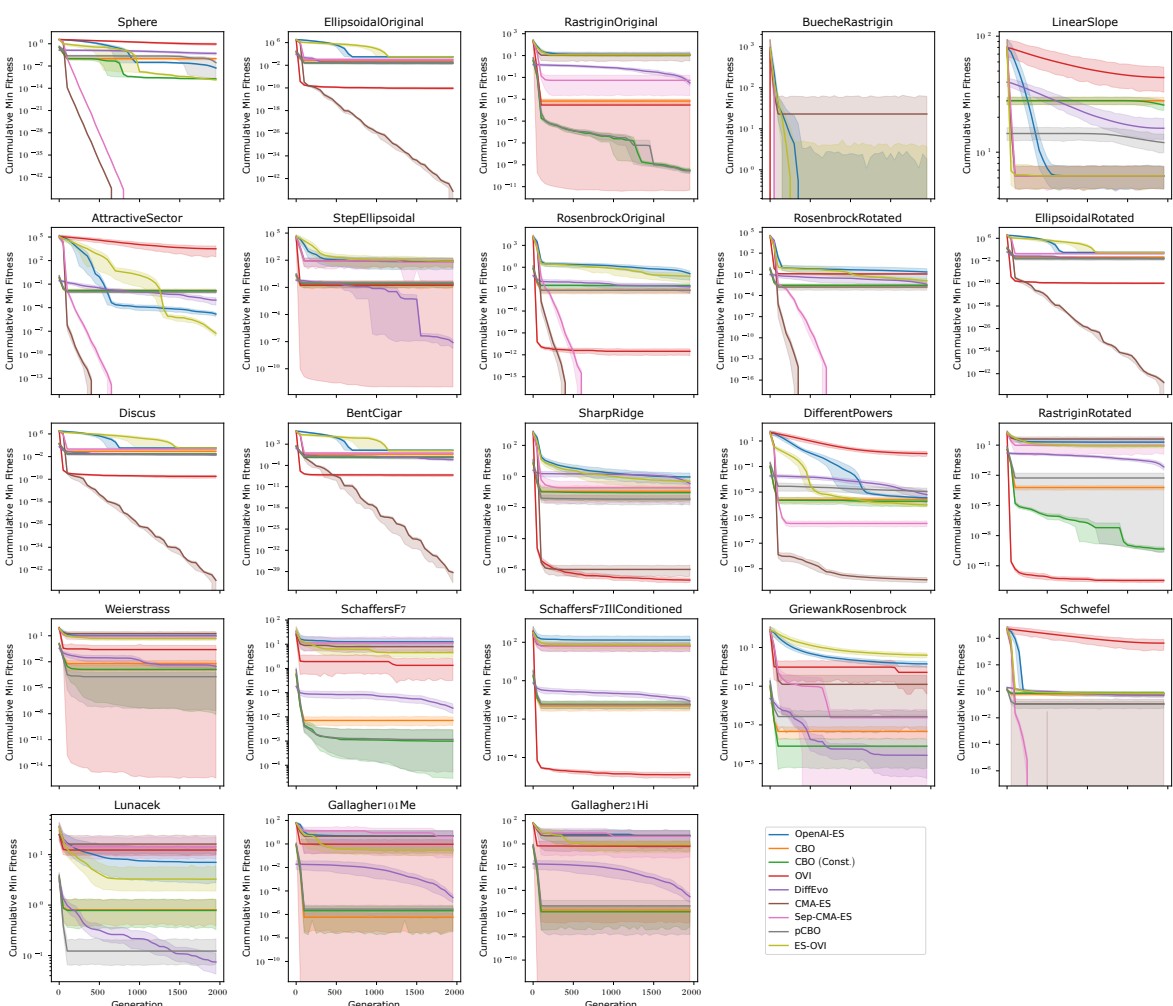

*Figure 9.* Results on BBOB tasks in 2D

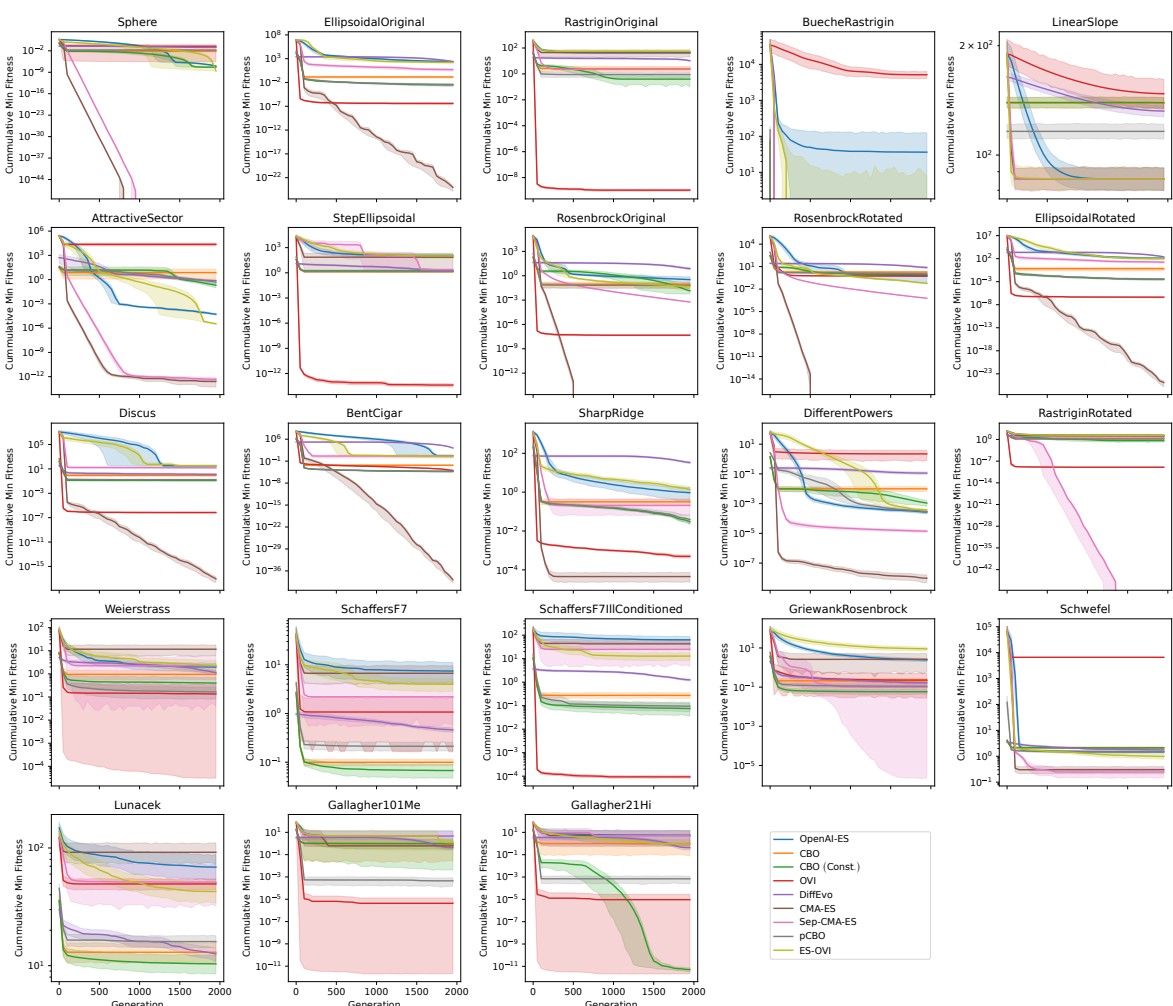

*Figure 10.* Results on BBOB tasks in 5D

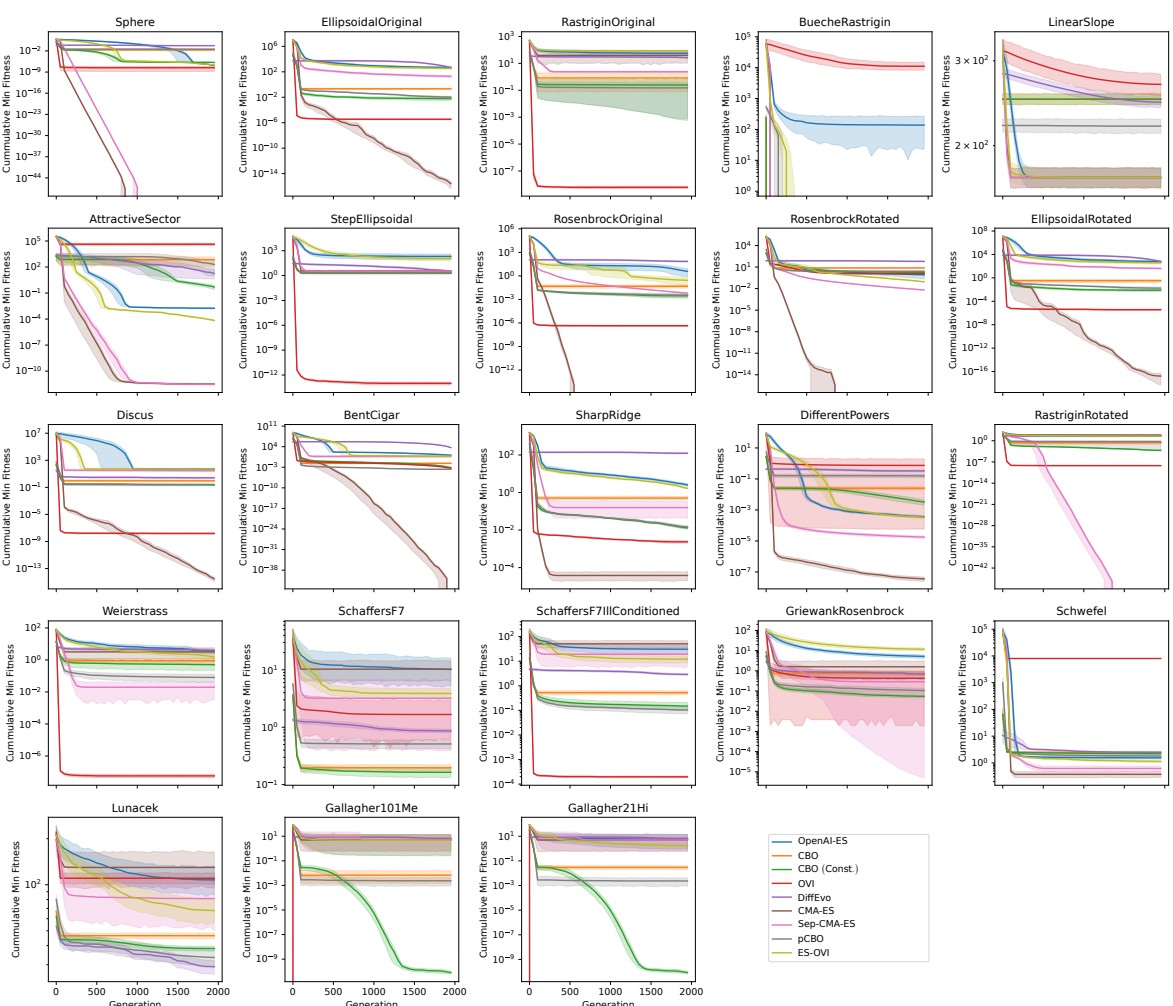

*Figure 11.* Results on BBOB tasks in 7D

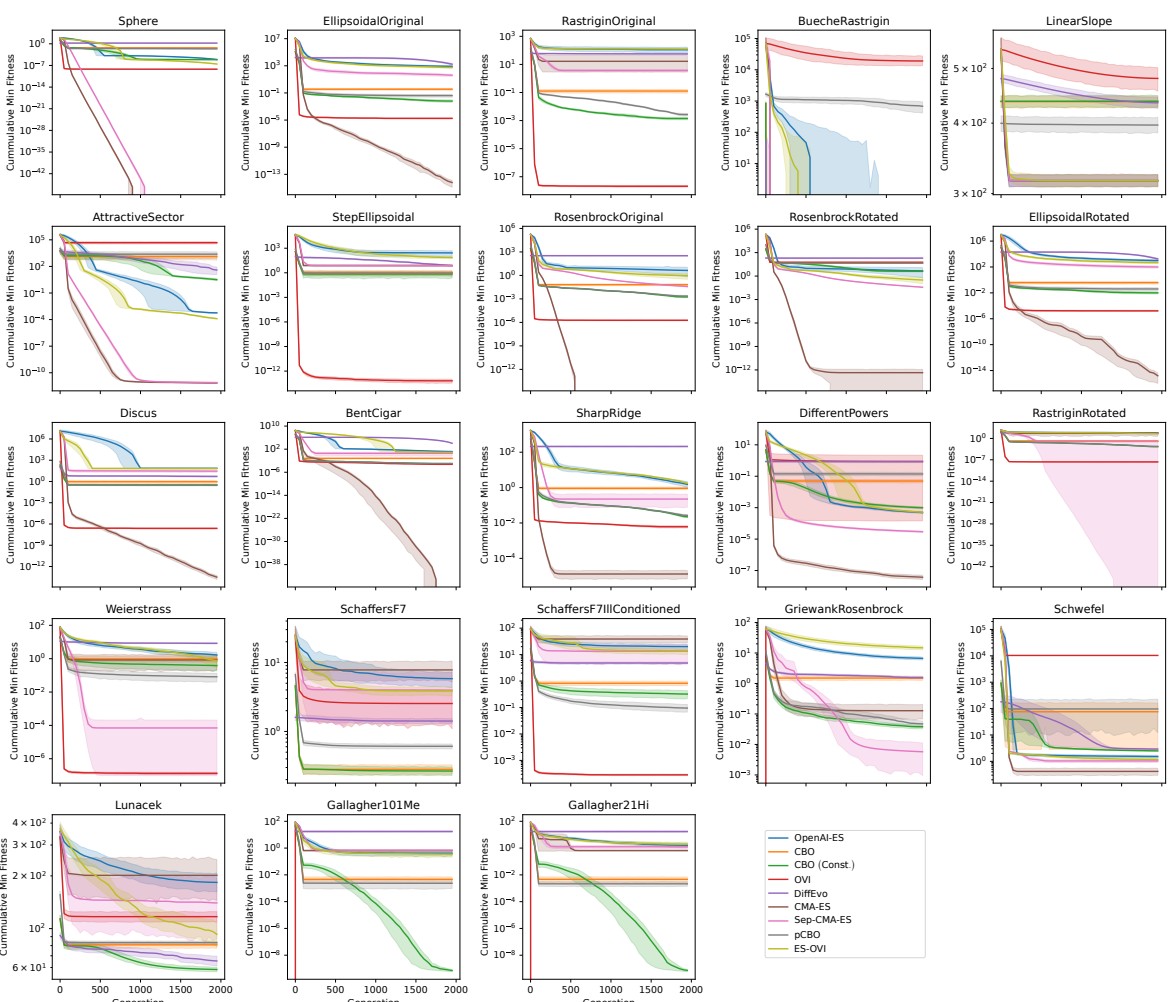

*Figure 12.* Results on BBOB tasks in 10D

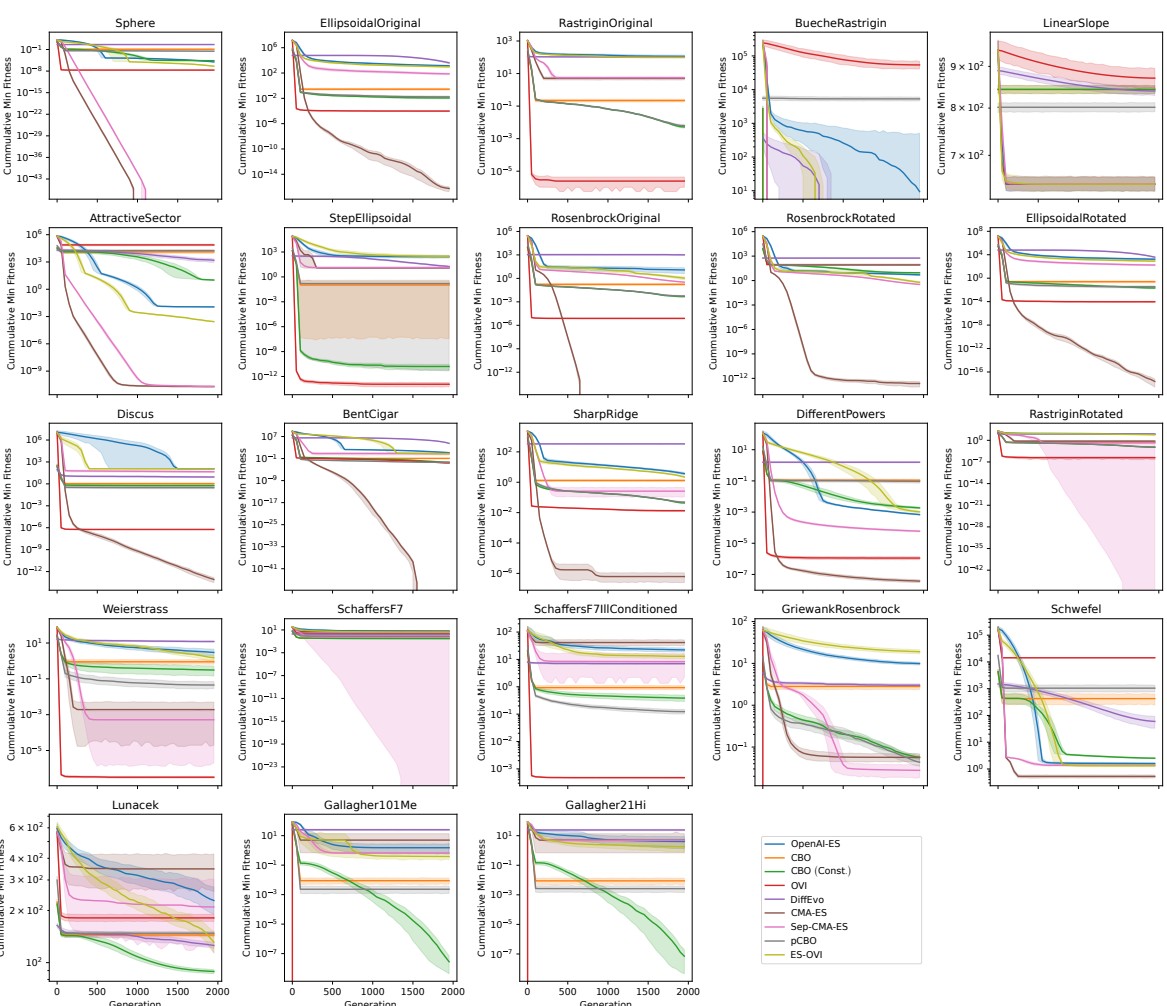

*Figure 13.* Results on BBOB tasks in 15D

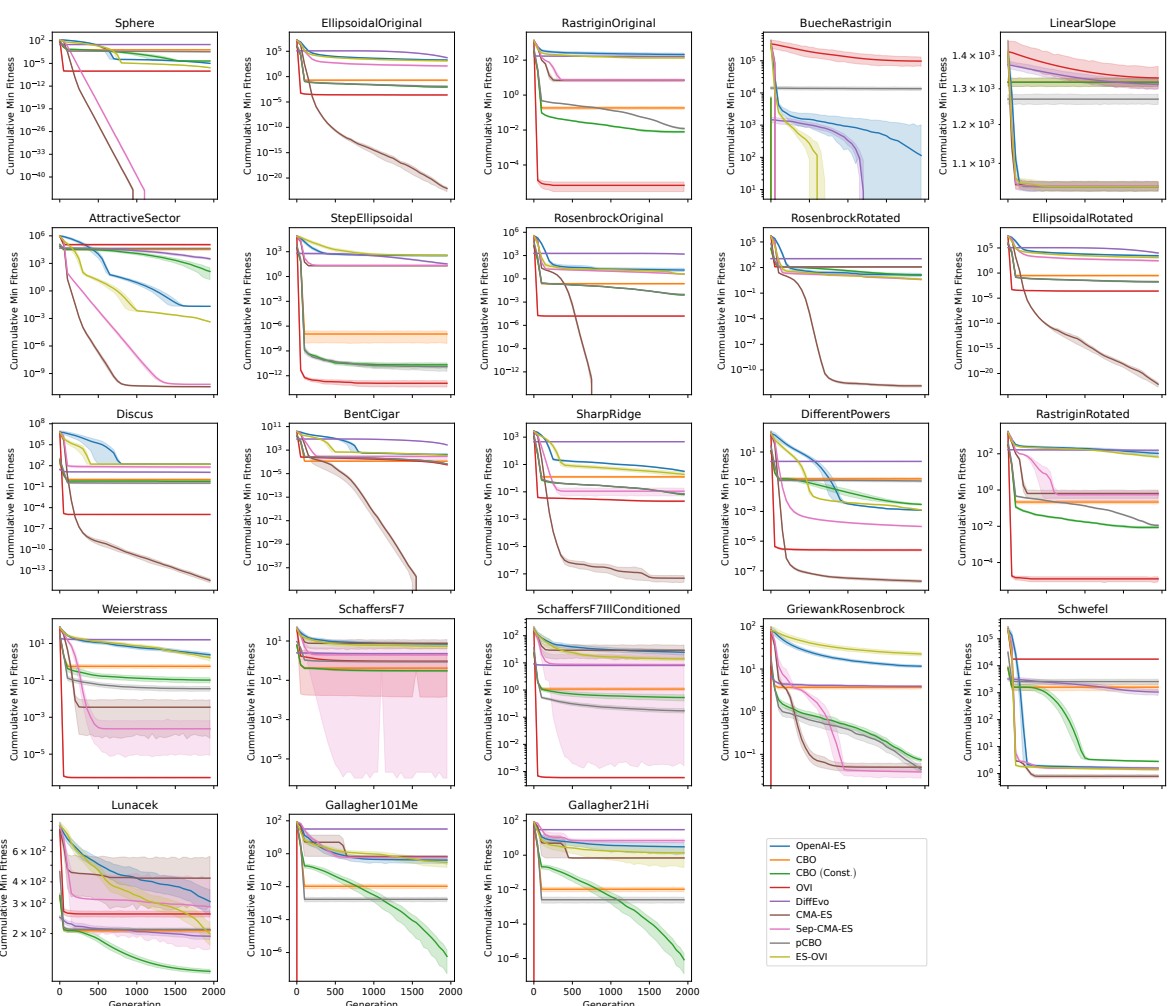

*Figure 14.* Results on BBOB tasks in 20D

