# OpenReview forum: "Bridging Spherical Black-Box Optimizers"
_ICML.cc/2026/Conference — ICML 2026 regular_

### Official Review · Reviewer_quPH · 2026-03-08

**Soundness:** 2
**Presentation:** 2
**Significance:** 2
**Originality:** 2
**Overall Recommendation:** 3
**Confidence:** 3

**Summary:**

The paper proposes a framework that unifies several methods from two families of black-box optimization algorithms, namely parametric approaches (particularly so-called evolution strategies) and non-parametric approaches (particularly particle-based methods). A common master update (MU) formulation is introduced, from which these algorithms can be derived, differing mainly in their mechanisms of fitness aggregation and the interaction scope among particles. Based on this formulation, several hybrid methods are proposed that interpolate between existing approaches and are reported to achieve superior performance compared to their constituent methods. The empirical evaluation is conducted on three types of benchmarks: the black-box optimization benchmark (BBOB), policy optimization for continuous control tasks, and large-language model (LLM) merging.

**Compliance With Llm Reviewing Policy:**

Affirmed.

**Final Justification:**

After reconsideration, I decide to raise my score from 2 to 3. However, given the current state-of-the-art results in black-box optimization (especially in evolutionary computation), both in theory and practice, it is difficult for me to give a higher rating.

**Key Questions For Authors:**

1) A more detailed discussion of the Information-Geometric Optimization (IGO) framework should be provided, particularly to clarify how the proposed framework relates to or differs from this existing theoretical perspective.

2) Scalability experiments on BBOB benchmarks across increasing problem dimensions (from low to high-dimensional instances) should be included.

3) Strong state-of-the-art evolutionary methods for continuous control, especially LM-MA-ES and ASEBO, should be included as baselines.

4) State-of-the-art multimodal evolutionary algorithms should also be evaluated on the LLM merging task.

**Limitations:**

The paper does not provide a dedicated discussion of its limitations, particularly regarding the lack of theoretical rigor, the absence of truly large-scale experiments, and the limited comparison with strong baselines.

**Strengths And Weaknesses:**

Soundness: The master update formulation is shown to unify several black-box optimization algorithms, including parametric methods (stochastic smoothing (SS), natural evolution strategies (NES), and Optimization via Integration (OVI)) and non-parametric methods (Consensus Based Optimization (CBO), polarized CBO (pCBO), clustered CBO (cCBO), Consensus Hopping (CH), and Diffusion Evolution (DE)). However, there is an issue with the naming convention used in the paper. The authors refer to Evolution Strategies (ES), but the methods considered correspond to Natural Evolution Strategies (NES) rather than classical ES as understood in the evolutionary computation literature. Classical ES methods typically involve rank-based selection and recombination of selected individuals to generate new candidate solutions. The ES formulations used in the paper (SS and NES) are gradient estimation methods, which are closer to policy-gradient style zeroth-order optimization.

Presentation: The presentation is mostly clear and well structured, beginning with the unified framework before introducing the derived hybrid methods and empirical evaluations. However, although the paper claims a theoretical framework as its main contribution, there is little to no formal proof accompanying the proposed framework.

Significance: The benchmarks and evaluations in the paper are somewhat limited in scale and the results are mixed. For the BBOB problems, only low-dimensional instances are considered. In the BBOB benchmark, it is generally important to evaluate algorithms across increasing problem sizes, from low- to high-dimensional instances, in order to assess their scalability. It is therefore unclear whether the conclusions drawn from these small problems would still hold on larger ones. The so-called high-dimensional continuous control benchmarks on Brax environments involve roughly 1000 variables, which are not truly high-dimensional by the standards of deep reinforcement learning. Moreover, important state-of-the-art variants of ES/NES designed for policy optimization are not included as baselines (e.g., Limited-Memory Matrix Adaptation Evolution Strategy (LM-MA-ES) or Adaptive ES-Active Subspaces for Black-box Optimization (ASEBO)). Similarly, although the model merging task involves LLMs, the number of decision variables is only 33 (i.e., the weights for merging parameters from different network layers), which still corresponds to a low-dimensional optimization problem. In addition, no state-of-the-art multimodal evolutionary algorithms are included as competing baselines.


Originality: It is interesting to see the unification of algorithms from the two families of parametric and particle-based approaches. However, there have been several prior attempts to unify black-box optimization algorithms, particularly within the evolutionary computation literature. In particular, the Information-Geometric Optimization (IGO) framework provides a principled derivation of distribution-based optimization algorithms such as NES and CMA-ES through natural gradient updates on the manifold of search distributions. The paper does not discuss IGO or compare its proposed framework with this existing line of work. Also, the proposed master update formulation appears less rigorous than the theoretical foundations provided by IGO.

---

> ### Author Rebuttal · Authors · 2026-03-31
>
> We would like to thank the reviewer for their helpful comments!
>
> >Q1 Discussion of the IGO framework, clarify how the proposed framework relates to or differs from this existing theoretical perspective.
>
> We agree that IGO [1] deserves more discussion and contextualization.
>
> In short, we focus on the connection of different parametric (NES/OVI/GS) and non-parametric (CBO/DiffEvo) methods, while IGO focuses on the connection between different parametric methods. IGO includes NES, but does not cover OVI, CBO, or DiffEvo.
>
> More in detail:
>
> IGO starts from a parametric family of search distributions $(P_\theta)_{\theta \in \Theta}$ together with an objective $f$ to be minimized. Given a nonincreasing selection function $w$, define
>
> $$
> q_\theta(x) := \Pr_{x' \sim P_\theta}\big(f(x') < f(x)\big),
> \qquad
> W_\theta^f(x) := w\big(q_\theta(x)\big).
> $$
>
> Thus, $W_\theta^f(x)$ is a rank-based utility assigned to $x$ relative to the current search distribution $P_\theta$.
>
> At time $t$, the current parameter is $\theta_t$, and the transform is frozen at that value. The IGO flow is
>
> $$
> \frac{d\theta_t}{dt} =
> I(\theta_t)^{-1}
> \int
> W_{\theta_t}^f(x)\,
> \nabla_\theta \log
> P_\theta(x)
> |_{\theta=\theta_t} P^{\theta_t} (dx)
> $$
>
> where $\tilde{\nabla}_\theta$ is the natural gradient and $I(\theta_t)$ is the Fisher information matrix. In practice, IGO then approximates the expectation and quantile transform by sampling.
> Ranking samples by their objective values to obtain $\hat w_i$, we have the sampled update
>
> $$
> \theta_{t+\delta t} =
> \theta_t +
> \delta t
> I(\theta_t)^{-1}
> \sum_{i=1}^N
> \hat w_i
> \nabla_\theta \log P_\theta(x_i)\Big|_{\theta=\theta_t}.
> $$
>
> Hence, the IGO algorithm is obtained by Euler discretization of the IGO flow, with rank-based utilities replacing the exact quantile transform.
>
> This construction is closely related to NES. The main differences are that NES is introduced directly as a discrete natural-gradient method for the expected fitness under the current search distribution. In practice, NES then replaces raw fitness values by rank-based utilities through fitness shaping [2]. The resulting final updates are thus very similar between NES and IGO.
>
> In the Gaussian setting, the main specializations discussed in [1] are NES/xNES, CMA-ES, and CEM.
>
> The overlap with our contribution is thus limited. Both works discuss NES, but we focus on the simplest restricted setting, spherical Gaussian distributions, in which NES becomes equivalent to OpenAI-ES and gradient smoothing. By contrast, [1] is concerned with a broader information-geometric formulation and with the relation of this formulation to existing parametric optimizers.
>
> To our understanding, OVI does not fit into the IGO framework because of its log-sum-exp transform, and the same is true for the nonparametric methods considered in our work, such as DiffEvo and CBO variants. More generally, [1] does not attempt to unify parametric and nonparametric methods.
>
> To the best of our knowledge, the unifying master equation / table presented in our work is therefore novel, as is the identification of the two main associated design choices: aggregation and consensus.
>
> In the revised manuscript, we will properly acknowledge the contributions of [1] and better contextualize our own contributions relative to this prior work.
>
> > Q2: Scalability experiments on BBOB
>
> We have run the BBOB benchmark for N=5, N=7, N=10, N=15 and N=20, the results can be seen in file BBOB_higher_dimensionality.pdf in the anonymous repository under the link https://anonymous.4open.science/r/bridging_spherical_bbo_rebuttal-9053/
> In general, we find that the hybrid method performs well in mostly the same tasks in higher dimensionalities as in the 2D experiments presented in the submission. We will add these results to the paper
>
> > Q3/Q4: State-of-the-art experiments
>
> We do not claim sota results in our submission.
> As we focus on connections between optimizers, our experiments aim to compare the performance of the hybrid optimizers to the original methods, not to compare to specialized sota methods.
> We will revise the presentation to make this limitation more clear, and add a discussion of LM-MA-ES and ASEBO.
>
> >Natural Evolution Strategies vs classical ES
>
> We will state this more clearly in the paper.
>
> >although the paper claims a theoretical framework as its main contribution, there is little to no formal proof accompanying the proposed framework...the proposed master update formulation appears less rigorous than the theoretical foundations provided by IGO.
>
> All derivations are reported in Appendix A.
> We are willing to address any specific point that the reviewer feels can be improved upon.
>
> Finally, we would like to thank the reviewer for their helpful review.
>
> [1] Ollivier et al.  “Information-Geometric Optimization Algorithms: A Unifying Picture via Invariance Principles”, JMLR, 2017.
>
> [2] D. Wierstra et al. “Natural Evolution Strategies”, JMLR, 2014.

---

> > ### Author Rebuttal · Reviewer_quPH · 2026-04-03
> >
> > Thank you for the clarification and additional experiments. However, I still find the contribution of the paper quite limited. On the practical side, no new algorithm is proposed that demonstrates superior performance over SOTA black-box optimization methods, and the experiments are not truly large-scale. On the theoretical side, no rigorous proof is provided for convergence analysis. Therefore, I decide to retain my score.
> >
> > =====
> >
> > Regarding the reply from the authors, I update my comments as follows.
> >
> > 1) I would like to see more extensive scalability experiments based on the BBOB test suite (please consider the experiment configuration in https://coco-platform.org/testsuites/bbob/performancecomparisons.html), including similar performance comparison plots for the considered algorithms and the proposed hybrids. Comparing against true state-of-the-art black-box optimization methods would be important to better position the contribution. Without such comparisons, it remains unclear under what scenarios the proposed hybrids would be preferable in practice, especially when stronger alternatives already exist.
> >
> > 2) By "formal proof," I mean convergence analysis. I apologize for not stating this clearly in my original reviews. I was surprised that a paper positioned as theoretical work on optimization lacks a convergence proof.
> >
> >
> > =====
> >
> > After reconsideration, I decide to raise my score from 2 to 3. However, given the current state-of-the-art results in black-box optimization (especially in evolutionary computation), both in theory and practice, it is difficult for me to give a higher rating.

---

> > > ### Author Response · Authors · 2026-04-04
> > >
> > > We would like to thank the reviewer for their rebuttal acknowledgement.
> > >
> > > The goal of our work is not to set a new SOTA, it is to provide a better understanding of the relationship between different black box optimization methods.
> > >
> > > We believe that these connections are valuable on their own, as they help advance our understanding of different methods, but we also show that they can be practically utilized to control convergence characteristics and improve results in some tasks.
> > >
> > > Providing SOTA results thus does not advance the goal of our paper.
> > >
> > > >Q1 A more detailed discussion of the Information-Geometric Optimization (IGO) framework should be provided, particularly to clarify how the proposed framework relates to or differs from this existing theoretical perspective.
> > >
> > > In the rebuttal we have reviewed the IGO method, and shown that the overlap with the methods considered in this work is limited to the NES method.
> > >
> > > >Q2 Scalability experiments on BBOB benchmarks across increasing problem dimensions (from low to high-dimensional instances) should be included.
> > >
> > > We have run the requested experiments and provided results for different dimensionalities of the BBOB benchmark.
> > >
> > > We are surprised that, despite addressing 2 of 4 concerns, the other two being related to SOTA, the reviewer maintains their score, and instead added a new concern about convergence analyses which they did not previously mention.
> > >
> > > We would like to again ask the reviewer about their original review:
> > >
> > > > “However, although the paper claims a theoretical framework as its main contribution, there is little to no formal proof accompanying the proposed framework“. “The proposed master update formulation appears less rigorous than the theoretical foundations provided by IGO”. “The paper does not provide a dedicated discussion of its limitations, particularly regarding the lack of theoretical rigor…”
> > >
> > >
> > > We believe all connections are clearly laid out in the main text and appendix. It would be helpful if the reviewer could point out which parts are problematic or not rigorous, that justify the current clearly negative recommendation
> > >
> > > ### =====
> > > Regarding the update provided by the reviewer:
> > >
> > > Thank you for the update.
> > >
> > > >BBOB test suite
> > >
> > > We do not see a noticeable benefit to using a different testsuite for comparisons to specialized SOTA methods, that have been (implicitly or explicitly) optimizing for performance on this very benchmark.
> > > We do not expect our methods to achieve SOTA results there, this is not our goal and was never claimed by us.
> > > Our abstract clearly states "demonstrating that the hybrid methods can outperform their constituent algorithms".
> > >
> > > > I was surprised that a paper positioned as theoretical work on optimization lacks a convergence proof.
> > >
> > > This was not the goal of our work either. If the wording of "theoretical framework" in the abstract evoked this expectation, we would be more than happy to change it to "unifying framework".
> > >
> > > Overall, we are surprised by the insistence on the clearly negative rating and seeming change in concerns during the rebuttal process, after we thought to have addressed the reviewer's main concerns in the initial rebuttal.

---

### Official Review · Reviewer_Y1DH · 2026-03-11

**Soundness:** 2
**Presentation:** 3
**Significance:** 2
**Originality:** 2
**Overall Recommendation:** 4
**Confidence:** 2

**Summary:**

This paper studies connections between several black-box optimization methods used when gradient information is unavailable. It proposes a unified theoretical framework showing that they can be interpreted as instances of a common master update equation. Within this framework, the methods mainly differ in two design choices: fitness aggregation, which controls the sharpness preference of the optimizer, and consensus scope, which determines the level of interaction among candidate solutions and affects the ability to discover multiple optima.

Leveraging this unified view, the authors propose hybrid optimizers that interpolate between parametric distribution-based methods (ES/OVI) and particle-based approaches (CBO). In particular, the paper introduces ES-OVI hybrids and CBO-OVI hybrids, which aim to combine high-dimensional efficiency with multimodal exploration capabilities.

The proposed methods are evaluated on black-box optimization benchmarks, continuous control tasks, and language model merging tasks under limited evaluation budgets. Results suggest that the hybrid methods can outperform their constituent algorithms.

**Compliance With Llm Reviewing Policy:**

Affirmed.

**Key Questions For Authors:**

1. Generality of the spherical assumption: the proposed unifying framework relies on spherical search distributions. Could the framework be extended to other type of search distributions? If not, how limiting is this assumption practice?
2. Computational overhead of hybrid methods: what is the runtime and per-iteration computational cost of the hybrid optimizers compared to the base algorithms (ES, OVI, CBO)?

**Limitations:**

- Restricted theoretical scope. The unification relies on spherical search distributions, which may limit applicability to problems where anisotropic exploration is necessary.

**Strengths And Weaknesses:**

**Strengths**

- Clear conceptual contribution. The paper proposes a unified framework that connects several major black-box optimization methods that are typically studied separately.

- Insightful perspective on optimizer design. The analysis identifies two key design axes (fitness aggregation and consensus scope), providing useful intuition about the behavior of different optimizers.

- Novel hybrid methods. The proposed hybrids (ES-OVI and CBO-OVI variants) leverage the theoretical insights to combine strengths of existing approaches.

- Empirical validation across multiple domains. The evaluation includes standard BBO benchmarks and applications such as locomotion and language model merging.

**Weaknesses**

- The theoretical unification relies on spherical search distributions, which is a restrictive assumption.

- Unclear practical benefit of the theoretical unification. It is not fully clear how much the theoretical perspective directly informs the design of the proposed hybrid algorithms or improves optimization performance beyond empirical tuning.

---

> ### Author Rebuttal · Authors · 2026-03-31
>
> > Q1: Generality of the spherical assumption: the proposed unifying framework relies on spherical search distributions. Could the framework be extended to other type of search distributions? If not, how limiting is this assumption practice?
>
> We believe extending it to other fixed search distributions would be possible, however, extending it to other methods which adapt the search distribution based on the data is non-trivial. Perhaps most prominently CMA-ES is not covered by our setting, as it adapts the covariance matrix dependent on the data. While CMA-ES is popular in practice, spherical search distributions are also used, especially in high-dimensional problems such as LLM fine-tuning (Qiu et al. 2025) or robotics (Al-Hafez and Steil 2021)
>
> > Q2: Computational overhead of hybrid methods: what is the runtime and per-iteration computational cost of the hybrid optimizers compared to the base algorithms (ES, OVI, CBO)?
>
> The total runtime for 1000 iterations with population size 256 and different problem dimensions is shown in the table below, averaged across 10 trials each.
> We are using a vectorized implementation in jax, thus for most methods the dimensionality does not have a large impact on the runtime.
>
> |        | OpenAI-ES | OVI  | ESOVI | CMA-ES | Sep-CMA-ES | CBO  | DiffEvo | AdaPol |
> |--------|-----------|------|-------|--------|------------|------|---------|--------|
> | D=5    | 4.0s      | 4.1s | 5.8s  | 9.9s   | 7.7s       | 5.9s | 4.9s    | 13.8s  |
> | D=50   | 4.0s      | 4.1s | 5.8s  | 10.3s  | 7.7s       | 6.0s | 5.0s    | 13.8s  |
> | D=1000 | 4.1s      | 4.2s | 6.1s  | 14.5s  | 7.8s       | 6.2s | 5.1s    | 13.8s  |
> | D=5000 | 4.2s      | 4.3s | 6.1s  | 149s   | 8.0s       | 6.5s | 5.7s    | 14.0s  |
>
> While our proposed hybrid methods (ESOVI and AdaPol) are slightly slower, the overhead should be negligible in practice, where function evaluation typically dominates the computational cost.
>
>
> (Al-Hafez and Steil) “Redundancy Resolution as Action Bias in Policy Search for Robotic Manipulation”, CORL 2021
> (Qiu et al.) “Evolution Strategies at Scale: LLM Fine-Tuning Beyond Reinforcement Learning”, arXiv, 2025

---

### Official Review · Reviewer_Frdo · 2026-03-12

**Soundness:** 4
**Presentation:** 3
**Significance:** 3
**Originality:** 3
**Overall Recommendation:** 5
**Confidence:** 4

**Summary:**

This paper presents a unified mathematical framework that bridges the gap between two separate families of spherical BBO methods: parametric distribution-update methods (such as ES and OVI) and nonparametric particle-based methods (such as CBO and its variants). By introducing a master update equation, the authors demonstrate that these algorithms differ primarily in their fitness aggregation strategies (which control the preference for optimum sharpness) and interaction scopes (which control the modality).

Based on this framework, the paper provides rigorous theoretical insights, formally proving that CH is equivalent to OVI, and that DE can be viewed as a time-varying pCBO. Then, the authors propose novel hybrid optimizers: ES-OVI, which interpolates gradients to explicitly control the flatness / robustness of the converged solution; and AdaPol / SchedPol, which dynamically schedule the attraction parameter to combine OVI's convergence capability with cCBO's multimodal exploration. The proposed methods are empirically evaluated on BBOB functions, Brax continuous control tasks, and an LLM merging task under limited evaluation budgets.

**Compliance With Llm Reviewing Policy:**

Affirmed.

**Final Justification:**

I think the authors' response clearly clarifies the scope of this paper and addresses my concerns. Therefore I remain my positive score. The field of BBO still lacks some principles with theoretical rigor, like PPO formulation in RL. I believe this paper opens a new line to bridge this gap.

**Key Questions For Authors:**

- Given the unified view, is there a principled way or a meta-level method to select the algorithmic components (e.g., the form of $\Psi(F)$, the interaction matrix $K$) or crucial hyperparameters based on the observed properties of an unknown target function during the early stages of optimization? Can practitioners leverage this framework to avoid exhaustive grid search in real-world BBO applications? Or you can give a brief summary in the abstract / introduction.

- In the LLM merging experiment, the dimensionality is d = 33. Could you provide experiments on more dimensionalities (d > 100 or 300)? The dimensionality of 33 usually does not fit the requirements of the real-world tasks.

**Limitations:**

The authors do not explicitly mention the limitations of this paper. I think one major limitation is that this paper is built upon continuous optimization, while input with discrete or constrained space is not discussed. This can be an interesting future work.

**Strengths And Weaknesses:**

## Strengths

- This paper is technically sound. The proposed master update framework is elegant and rigorous. The empirical evaluation is well-designed, covering synthetic BBOB functions, RL control, and LLM merging. I also appreciate the authors' transparency to deliver experimental results, e.g., acknowledging "*All methods are unable to match the accuracy of the method proposed by Akiba et al. (2025) due to overfitting of the training dataset*" in Section 5.3.
- The manuscript is well-written. Though I am not so familiar with OVI and CBO, I can follow the authors' presentation.
- Unifying such BBO methods is a good contribution, pushing towards principles for BBO practitioners.
- This paper is also of originality. I never read a paper bridging the connection among the discussed methods.

## Weaknesses

- Although the unified framework provides instantiation of some core issues of BBO methods (e.g., exploration / exploitation) and shows that different methods own different inductive biases, it remains unknown how practitioners can leverage specific methods when dealing with real-world BBO tasks. I suggest providing some principals, especially when the user does not know the landscape of the objective function.
- The authors report best performing one after grid search on hyperparameters. It is unclear how to select hyperparameters for real-world budget-aware use.
- Typos, mainly in the appendix:
   - line 47, "black box" -> "black-box";
   - line 215, "high dimensional" -> "high-dimensional";
   - line 413, "high dimensional" -> "high-dimensional";
   - line 444, "none which" -> "none of which";
   - line 554, "Braun et al. (2025) has investigated" -> "Braun et al. (2025) have investigated";
   - line 562, "Carrillo et al. (2021) introduces" -> "Carrillo et al. (2021) introduce";
   - line 758, "proportionate" -> "proportionally";
   - line 782, "in Table 2 We tune" -> "in Table 2. We tune";
   - line 814, "Table 3 For the" -> "Table 3. For the";
   - line 880, "Hyper-paramter" -> "Hyper-parameter";
   - line 901, "and thus use TIES" -> "and thus we use TIES".

---

> ### Author Rebuttal · Authors · 2026-03-31
>
> We would like to thank the reviewer for their helpful review.
>
> >Given the unified view, is there a principled way or a meta-level method to select the algorithmic components (e.g., the form of , the interaction matrix ) or crucial hyperparameters based on the observed properties of an unknown target function during the early stages of optimization? Can practitioners leverage this framework to avoid exhaustive grid search in real-world BBO applications?
>
> This is a very good suggestion. It could be possible to guess whether a given problem is multimodal or unimodal from a few initial evaluations, allowing us to choose the interaction matrix. Our proposed AdaPol method already does this indirectly by using a bandit setting, although a more direct utilization of the data may be possible.
> On the other hand, whether convergence to a flatter or sharper minimum is desired seems difficult to predict in the early training stages, as local flatness may vary highly throughout training.
> We will add a discussion about this to the paper, however, we believe a fully automated solution is better left to future work.
>
>
> >In the LLM merging experiment, the dimensionality is d = 33. Could you provide experiments on more dimensionalities (d > 100 or 300)? The dimensionality of 33 usually does not fit the requirements of the real-world tasks.
>
> While we agree that in principle a higher dimensionality could allow for a better performance in the merging setting, preliminary experiments showed it to result in stronger overfitting to the limited training dataset, thereby decreasing the test accuracy.
> We would like to note that we also evaluated the combined CBO-OVI method in the brax tasks with d~1000, where they performed better than standard CBO.
>
> We further added additional experiments on different dimensionalities of the BBOB tasks for the rebuttal here:
> https://anonymous.4open.science/r/bridging_spherical_bbo_rebuttal-9053/
>
> >Typos
>
> We thank the reviewer for reporting the typos, we will fix them for the final version of the paper!

---

> > ### Author Rebuttal · Reviewer_Frdo · 2026-04-01
> >
> > Thank you for the detailed response. I would like to maintain my positive score. The authors' response addresses my concerns. For a future revised version, I suggest: 1) providing discussion/insights to more realistic settings instead of the spherical distributions, as also mentioned by Reviewer Y1DH; 2) making the implementation open-sourced.

---

> > > ### Author Response · Authors · 2026-04-05
> > >
> > > We would like to thank the reviewer for their rebuttal acknowledgements and are glad that we could address their concerns.
> > >
> > > We plan to make our implementation of the used algorithms public after acceptance, and will add the discussion of the spherical setting. We would like to note that, while a limitation, we believe the spherical setting is also practically useful, as discussed in the response to reviewer Y1DH.

---

### Official Review · Reviewer_kLUy · 2026-03-12

**Soundness:** 3
**Presentation:** 3
**Significance:** 4
**Originality:** 4
**Overall Recommendation:** 4
**Confidence:** 3

**Summary:**

This paper addresses the gap between two traditionally distinct families of black-box optimizers: parametric methods (e.g., Evolution Strategies (ES) and Optimization via Integration (OVI)) and nonparametric particle methods (e.g., Consensus-Based Optimization (CBO)). The authors propose a unified Master Update (MU) framework, demonstrating that most existing BBO algorithms are special cases of this single equation. Leveraging these insights, the paper introduces hybrid optimizers such as ES-OVI, which allows explicit control over the preference for flat minima to improve robustness, and AdaPol/SchedPol, which combines the high-dimensional efficiency of OVI with the multimodal capabilities of CBO.

**Compliance With Llm Reviewing Policy:**

Affirmed.

**Final Justification:**

I would like to thank the authors for their detailed responses. I have decided to maintain my current score. My primary concerns remain regarding the fact that the failure modes in certain benchmarks are not yet fully understood or explained. Nevertheless, I believe the overall contribution of the work is valuable to the community.

**Key Questions For Authors:**

* Why do the hybrid methods fail to significantly outperform traditional baselines in final fitness across several benchmarks, despite better initial convergence?
* How sensitive are the results to the new hyperparameters (α and adaptive schedules), and are there general heuristics for tuning them?
* Beyond formal unification, are there rigorous convergence rate guarantees for the proposed hybrid variants in non-convex settings?

**Limitations:**

The authors should discuss the practical limitations of the hybrid methods—specifically the increased hyperparameter tuning overhead.

**Strengths And Weaknesses:**

## Strengths
* The paper provides an elegant and comprehensive framework (MU) that bridges previously disconnected research streams, significantly simplifying the conceptual landscape of BBO.
* The authors demonstrate the scalability of their approach by moving beyond simple 2D benchmarks to high-dimensional problems (d≈1000 in Brax) and real-world LLM parameter-space optimization.
* The proposed hybrid methods (like ES-OVI) utilize the same samples and evaluations as their constituent parts, incurring no additional computational overhead per iteration.

## Weaknesses
* **Marginal performance gains in some benchmarks**: In several experiments (e.g., certain Brax and BBOB tasks), the hybrid methods either perform similarly to traditional ES/CMA-ES or show superior initial convergence but fail to surpass the final results of existing baselines.
* **Increased hyperparameter sensitivity**: While the framework offers more control, it introduces new hyperparameters (such as the interpolation coefficient α or adaptive schedules). In truly "black-box" scenarios where the landscape is unknown, tuning these additional parameters may increase the user's overhead.
* **Limited theoretical proofs:** : The paper excels at "formal unification" (showing algorithms can be written in the same form), but it lacks rigorous mathematical proofs regarding the convergence rates or global optimality guarantees for the newly proposed hybrid variants in non-convex settings.

---

> ### Author Rebuttal · Authors · 2026-03-31
>
> We would like to thank the reviewer for their helpful review!
>
> > Q2: How sensitive are the results to the new hyperparameters (α and adaptive schedules), and are there general heuristics for tuning them?
>
> While we do not have an ablation for the adaptive schedules, we did not need to do any significant tuning on the new hyper-parameters for combined OVI-CBO methods, specifically the duration over which to average the success rate of each method and how many particles to assign to each method initially.
> For ES-OVI we show the performance for different values in Figure 6 on brax tasks. While α definitely has a strong impact on the performance, we find that simply setting α=0.5 works reasonably well across tasks. Otherwise we would recommend using more ES-like behavior when there are many undesirable sharp optima, but lean towards OVI-like behavior if sharp minima are acceptable in the specific problem we are optimizing.
> We will add a discussion to the paper!
>
> > Q1: Why do the hybrid methods fail to significantly outperform traditional baselines in final fitness across several benchmarks, despite better initial convergence?
>
> We assume this mainly refers to the hybrid ES-OVI in the Hopper, Reacher, and Halfcheetah tasks.
> In these tasks, the candidates represent the hidden parameters of a neural network controller with tanh activations. This results in a complex loss landscape with many equivalent candidates, making it hard to analyse and give a precise reason for this failure.
> We note that this failure mode does not appear consistently in the more diverse BBOB problems.
>
> > Q3: Beyond formal unification, are there rigorous convergence rate guarantees for the proposed hybrid variants in non-convex settings?
>
> We do not provide any convergence rates.
>
> >Limitations:
>
> We will discuss the additional hyperparameters and their choice in the paper.
>
> Thank you for the review!

---

> > ### Author Rebuttal · Reviewer_kLUy · 2026-03-31
> >
> > I would like to thank the authors for their detailed responses. I have decided to maintain my current score. My primary concerns remain regarding the fact that the failure modes in certain benchmarks are not yet fully understood or explained. Nevertheless, I believe the overall contribution of the work is valuable to the community.

---

### Decision · Program_Chairs · 2026-04-30

**Decision:**

Accept (regular)

**Comment:**

The paper investigates the bridging of spherical black-box optimizers, a topic that the reviewers generally agree is interesting and potentially valuable to the community. The author response mostly addresses the reviewer comments. The authors should incorporate the new experiments, as well as improve writing in the camera ready to address the reviewer feedback.